# Reasoning Through Execution: Unifying Process and Outcome Rewards for Code Generation

Zhuohao Yu [1] [*]   Weizheng Gu [1] [*]   Yidong Wang [1]   Xingru Jiang [1]   Zhengran Zeng [1]
Jindong Wang [2]   Wei Ye [1]   Shikun Zhang [1]

## Abstract

Large Language Models excel at code generation yet struggle with complex programming tasks that demand sophisticated reasoning. To bridge this gap, traditional process supervision relies on learned reward models requiring costly training data and suffering from reward misalignment, while outcome supervision fails for complex tasks needing coordinated intermediate steps. We introduce Outcome Refining Process Supervision, which unifies process and outcome supervision by leveraging executable verification: a tree-structured search framework generates strategic alternatives, profiles execution metrics, and scores candidates via self-critique mechanisms that integrate runtime feedback with reasoning. Experiments across 5 models and 3 benchmarks show consistent gains, with **26.9%** higher correctness and **42.2%** improved code efficiency. The results demonstrate that ORPS enables LLMs to overcome local optima in code generation, suggesting a promising direction for combining verifiable outcomes with structured reasoning to tackle complex challenges. We open-source at: https://github.com/zhuohaoyu/ORPS

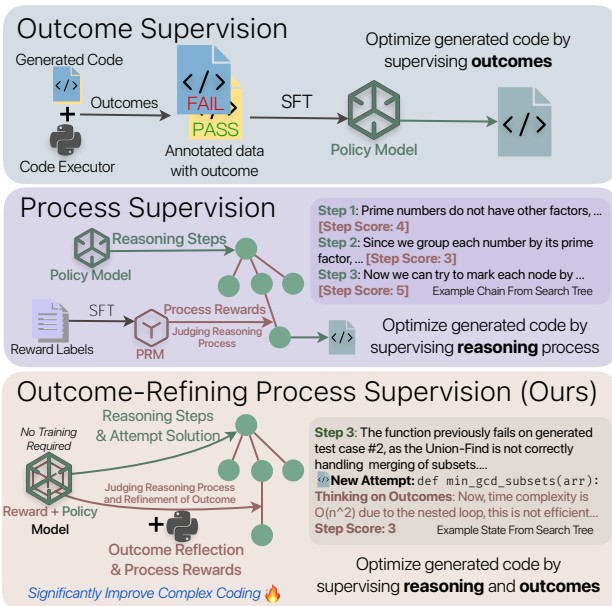

Figure 1: Comparison of outcome and process supervision.

## 1. Introduction

Large Language Models (LLMs) have revolutionized code generation through their ability to synthesize programs from natural language specifications (Brown et al., 2020; Guo et al., 2024). However, complex programming tasks requiring multi-step algorithmic reasoning—such as implementing dynamic programming solutions or optimizing parallel computation patterns—remain challenging (Jiang et al., 2024b; Jimenez et al., 2023). These limitations persist even in state-of-the-art models (OpenAI, 2023; Touvron et al., 2023a), revealing a critical gap in current supervision paradigms.

As shown in Figure 1, traditional approaches follow two main paradigms: *outcome supervision*, which evaluates only final outcome quality (Chen et al., 2021b), and *process supervision*, which guides intermediate steps using learned Process Reward Models (PRMs) with search algorithms (Lightman et al., 2023). While PRMs have shown success in mathematical reasoning (Wang et al., 2024c; Chen et al., 2024a), their application to code generation faces fundamental challenges: (1) PRMs require expensive human annotations or distillations of other models on intermediate steps to train (Wang et al., 2024b); (2) Learned rewards can suffer from *hallucination* (misjudging invalid steps as correct) (Huang et al., 2023; Stechly et al., 2024) and *reward hacking* (exploiting superficial patterns to maximize scores) (Skalse et al., 2022); (3) Code-specific PRMs are scarce, and math-focused PRMs may not suit program-

[1]Peking University, Beijing, China [2]William & Mary, VA, USA. Correspondence to: Wei Ye <wye@pku.edu.cn>.

*Proceedings of the 42nd International Conference on Machine Learning*, Vancouver, Canada. PMLR 267, 2025. Copyright 2025 by the author(s).

ming's structured logic. More fundamentally, our work questions the necessity of *separately trained* PRMs when LLMs' intrinsic reasoning capabilities can be effectively guided by verifiable outcomes.

Code generation offers a unique advantage through *concrete, verifiable signals*: code can be executed throughout development, providing objective feedback on correctness and performance (Zhang et al., 2023; Shinn et al., 2024). However, existing execution-feedback methods like Reflexion (Shinn et al., 2024), LDB (Zhong et al., 2024), Self-Repair (Olausson et al., 2023), and REx (Tang et al., 2024a) primarily focus on *local code repair*, missing opportunities to explore fundamentally different algorithmic strategies.

We propose **Outcome-Refining Process Supervision**, which treats the *reasoning about outcome refinement itself* as the process to be supervised. Unlike repair-focused methods that incrementally fix code blocks, ORPS operates at a *higher abstraction level*, guiding LLMs to reason about overall *solution strategies*. Through tree-structured exploration, our framework maintains multiple reasoning trajectories simultaneously, enabling models to explore different algorithmic approaches when initial attempts prove suboptimal (e.g., switching from brute-force to divide-and-conquer approaches), rather than being trapped in local optima (e.g., a brute-force solution that passes the test but is inefficient).

Our key insight is that LLMs' intrinsic reasoning and self-critique capabilities, when anchored by verifiable execution feedback, can generate high-quality process rewards, *eliminating the need for specially trained PRMs*. Execution outcomes and performance metrics provide objective grounding, while self-critique offers strategic guidance. This creates a powerful, inference-only feedback loop where reasoning chains and code implementations co-evolve: execution failures prompt strategic re-analysis, while refined algorithmic insights lead to better implementations.

ORPS is the first *inference-only* process supervision framework for code generation that systematically explores algorithmic strategies without PRM training. Experiments across 5 models and 3 benchmarks reveal:

- **Strategic Reasoning Over Model Scale:** Providing sufficient reasoning space for algorithmic exploration is more crucial than model size; ORPS enables smaller models to achieve high success rates, often outperforming larger counterparts using conventional methods.

- **Effective PRM Elimination:** Combining execution feedback with self-critique creates superior verification mechanism compared to learned PRMs, eliminating costly training while leveraging LLMs' inherent capabilities.

- **Algorithmic Improvement Beyond Local Optima:** ORPS consistently improves correctness and solution

efficiency by exploring superior algorithmic strategies rather than just performing local fixes.

Our key contributions include:

- We propose ORPS: a novel inference-only framework that unifies outcome and process supervision by guiding LLMs to reason about *solution strategies at a higher abstraction level*, moving beyond local code repair.

- We demonstrate that combining verifiable execution feedback with self-critique effectively *eliminates the need for specially trained PRMs*, outperforming them while reducing overhead.

- ORPS achieves substantial improvements across diverse benchmarks: an average Pass@1 improvement of 26.9% across datasets and models, while reducing running time by 42.2% on average.

## 2. Related Work

### 2.1. Outcome Supervision vs Process Supervision

**Outcome Supervision** in language models traditionally evaluates and optimizes LLM outputs through three primary paradigms: (1) *Open-ended generation* tasks (e.g. instruction following) assessed via text similarity metrics with reference answers (Zhang et al., 2019) or LLM-as-a-judge scoring (Zheng et al., 2023; Wang et al., 2023c; Yu et al., 2024a); (2) *Constrained-output tasks* (multiple-choice, math problems) judged by exact answer matching (Brown et al., 2020; Hendrycks et al., 2020) of answer keys or solutions; and (3) *Code generation* where correctness depends on test case execution with generated programs (Chen et al., 2021b; Austin et al., 2021). These outcomes are then used to curate the data (e.g., selecting higher-quality conversations or code solutions) for further training of the model (Liu et al., 2023b). Such approaches share a critical limitation: they ignore the reasoning process that produced the output (Uesato et al., 2022). This proves particularly problematic for complex tasks where optimal solutions require coordinated intermediate steps (Lightman et al., 2023).

**Process Supervision** addresses this gap by optimizing *intermediate reasoning trajectories* through specially trained *Process Reward Models (PRMs)* that score each intermediate step (Uesato et al., 2022; Lightman et al., 2023). PRMs have proven particularly effective in domains requiring complex reasoning, such as math problems, where they guide search or sampling algorithms toward better reasoning trajectories and solutions (Wang et al., 2024c; Chen et al., 2024a; Wang et al., 2024b). While outcome supervision gives $R_{\text{outcome}}(y) = \mathbf{1}[\text{correct}]$, process supervision gives step-wise reward signals $R_{\text{process}} = \sum_{t=1}^{T} \text{PRM}(s_t|s_{1:t-1})$.

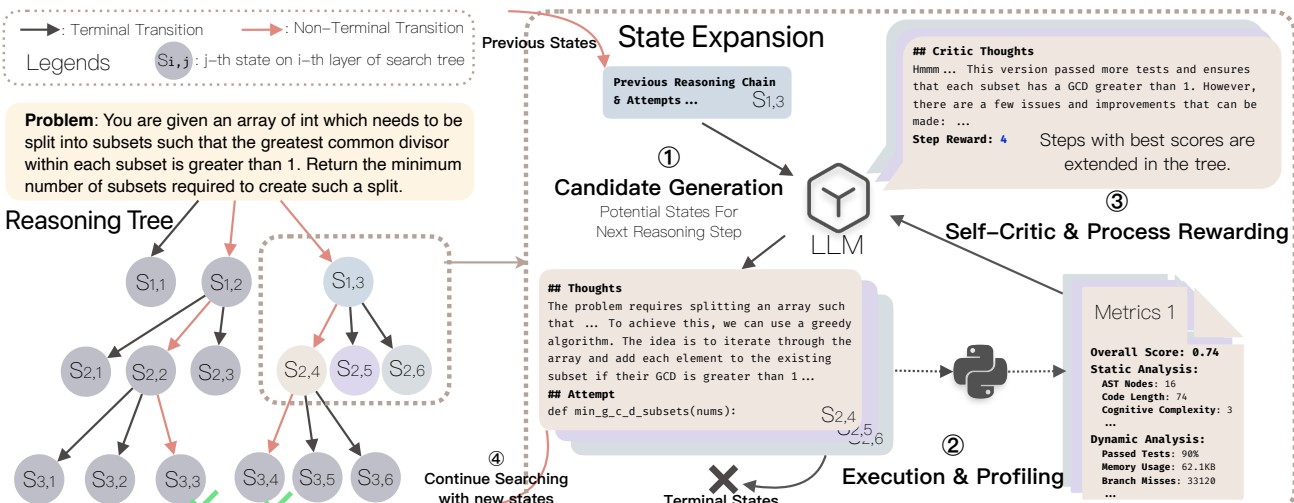

Figure 2: **Outcome-Refining Process Supervision** framework overview. A language model serves as both *programmer* and *critic* in a step-by-step reasoning process. Through beam search, the framework maintains multiple solution trajectories, where each state contains **reasoning chains**, **code implementations**, and **step reward**.

This approach has been predominantly used in math reasoning tasks(Luo et al., 2024; Jiang et al., 2024a), but with limitations: 1) The requirement for dense human annotations to train reliable PRMs makes the approach expensive and time-consuming (Lightman et al., 2023). 2) The generalization capability of PRMs is often limited, as reasoning patterns can vary significantly across different tasks and domains. 3) When serving as judges, LLMs may produce unreliable evaluations due to hallucination (Hu et al., 2024; Li et al., 2024), particularly for complex tasks (Thakur et al., 2024). Recent studies show that LLMs cannot reliably self-correct (Huang et al., 2023) or self-validate without external verification (Stechly et al., 2024). However, emerging work suggests that specially trained PRMs may be unnecessary when LLMs' intrinsic reasoning capabilities are effectively guided by verifiable outcomes (Chakraborty et al., 2025). These limitations motivate our approach of grounding process supervision in concrete, verifiable signals rather than learned judgments.

## 2.2. Execution-Driven Code Generation

Code generation is typically formulated as a sequence-to-sequence problem: given input specification $x$ (including natural language description and test cases), generate a program $y$ that correctly implements the required functionality (Jiang et al., 2024b). While most existing approaches treat this as a single-step generation process (Chen et al., 2021a), recent work has explored using execution feedback to guide code generation (Zhong et al., 2024; Zhang et al., 2023) or use CoT prompting to improve correctness (Shinn et al., 2024).

Self-Repair (Olausson et al., 2023) demonstrated that reasoning quality and self-repair capabilities correlate with code generation outcomes, establishing the importance of structured reasoning through repair trees. REx (Tang et al., 2024a) further explored code repair as an exploration-exploitation tradeoff. However, these execution-guided approaches primarily focus on *local code repair*—debugging specific functions or fixing syntax errors—missing opportunities to explore fundamentally different strategies.

Although these execution-guided approaches show promise, our experiments indicate they are insufficient for complex programming tasks that require deeper reasoning. While execution feedback is easy to measure, it alone provides little guidance on how to improve solutions that fail or how to make working solutions more efficient. More importantly, it offers no feedback during the intermediate stages of development, when early course corrections could prevent cascading errors.

Consider implementing an efficient sorting algorithm: a model might write code that passes all test cases but uses an $O(n^2)$ approach. Existing outcome supervision methods would mark this as a success, missing the opportunity to guide the model toward an optimal $O(n \log n)$ solution. Similarly, if the code fails, a sparse "fail" signal provides no insight into whether the error lies in the algorithmic approach, the implementation details, or edge case handling. These limitations highlight the need to rethink how to supervise the development of complex programs, where both theoretical understanding and practical implementation must evolve together—operating at a higher abstraction level than typical repair-focused methods.

# 3. Methodology

We propose a framework that unifies process supervision with outcome supervision by combing *reasoning*, *code implementation*, and *execution verification* into a single tree-structured search process.

When tackling coding tasks, particularly complex ones, it is challenging for a model to generate a fully correct solution on the first attempt. Instead, LLMs often produce imperfect yet heuristically valuable code, requiring iterative self-correction to eventually arrive at a correct implementation. Each iteration of code refinement can be considered a step in the problem-solving process.

As formalized in Algorithm 1 and illustrated in Figure 2, at step $t$, the node in a given search beam represents a state $s_t = (\mathcal{R}_t, C_t, F_t, \omega_t, K_t, \rho_t)$, where $\mathcal{R}_t$ denotes the current reasoning chain, $C_t$ the code implementation, $F_t$ the execution feedback, $\omega_t$ the outcome reward score, $K_t$ the self-critic reasoning and $\rho_t$ the process reward score.

The search progresses through three phases :

1. **Candidate Generation.** The LLM is leveraged to do reasoning on how to refine or alternative strategies and then attempt to generate corresponding code implementations.
2. **Execution & Profiling.** Each candidate code is executed and profiled on unit tests, to measure key performance metrics, such as correctness, efficiency, and code quality.
3. **Self-Critic & Process Rewarding.** The LLM is prompted to generate a texual self-critic considering reasoning chain and execution metrics, then gives a numerical process reward score which is used to guide searching.

Unlike linear CoT approaches that commit to a single trajectory, this tree structure enables *parallel exploration* of divergent strategies—for instance, maintaining both greedy and dynamic programming approaches for optimization problems until empirical feedback identifies the superior solution.

## 3.1. Candidate Generation

The candidate generation phase expands the search tree by producing diverse refinements. For each node $s_{t-1} = (\mathcal{R}_{t-1}, C_{t-1}, F_{t-1}, \omega_{t-1}, K_{t-1}, \rho_{t-1})$, the LLM generates $N$ successor candidates:

$$\{(r_t^{(j)}, c_t^{(j)})\}_{j=1}^N = \mathcal{M}(\mathcal{R}_{t-1}, C_{t-1}, F_{t-1}),$$

where $r_t^{(j)}$ represents incremental reasoning updates (e.g., 'Adjust loop termination to prevent off-by-one errors') or strategic pivots (e.g., 'Replace recursion with iteration to avoid stack overflow'). This dual-generation mechanism

---

**Algorithm 1** Outcome-Refining Process Supervision

**Input:** Problem $x$, Unit Tests $U$, Model $\mathcal{M}$, Beam size $K$, Steps $T$, Candidates $N$, Process Reward Weight $\alpha$, Outcome Reward Weight $\beta$
Initialize $\text{beam}_0 \leftarrow \{(x, \emptyset)\}$ {Start with problem description}
**for** step $t = 1$ **to** $T$ **do**
  $\text{paths} \leftarrow \emptyset$ {Initialize reasoning paths}
  **for** state $s$ in $\text{beam}_{t-1}$ **do**
    $\text{chain} \leftarrow s.\text{reasoning\_chain}$ {Copy current reasoning chain}
    $\text{candidates} \leftarrow \mathcal{M}_{\text{reason}}(x, \text{chain}, N)$ {Generate N pairs}
    **for** candidate in candidates **do**
      $\text{reasoning}, \text{code} \leftarrow$ Extract from candidate
      $\text{newchain} \leftarrow \text{chain} \oplus \text{reasoning} \oplus \text{code}$
      $\text{feedback}, \text{outcome\_rew} \leftarrow \text{execute\_and\_profile}(\text{code}, U)$
      $\text{newchain} \leftarrow \text{newchain} \oplus \text{feedback} \oplus \text{outcome\_rew}$
      $\text{crit}, \text{process\_rew} \leftarrow \mathcal{M}_{\text{critic}}(\text{newchain}, \text{code}, \text{feedback})$
      $\text{newchain} \leftarrow \text{newchain} \oplus \text{crit} \oplus \text{process\_rew}$
      $\text{step\_score} \leftarrow \alpha \times \text{process\_rew} + \beta \times \text{outcome\_rew}$
      $\text{paths} \leftarrow \text{paths} \cup \{(\text{newchain}, \text{step\_score})\}$
    **end for**
  **end for**
  $\text{beam}_t \leftarrow$ Select Top-$K$ paths with highest $\text{step\_score}$
  **if** Any path in $\text{beam}_t$ is complete **then**
    **break**
  **end if**
**end for**
**Return:** Best reasoning chain from $\text{beam}_T$

---

ensures *algorithmic diversity*—maintaining competing approaches until empirical feedback resolves ambiguities.

## 3.2. Execution & Profiling Outcome Rewards

We execute the model-generated code $\{c_t^{(j)}\}_{j=1}^N$ on a set of predefined Unit tests $U$ and execute them to obtain results. $U$ may be either generated by the LLM itself based on the problem or provided by the dataset. We will discuss the results under different scenarios in subsection 4.1.

The outcome reward $\omega_t^{(j)}$ combines *dynamic analysis* (Ball, 1999) and *static analysis* (Nielson et al., 1999) metrics into a unified score, grounding solution quality in both runtime behavior and structural properties. We compute:

$$\omega_t^{(j)} = \sum_{k=1}^M \beta_k \cdot \text{normalize}(m_k^{(j)}),$$

We have correctness, execution time, CPU instruction count, page faults as dynamic analysis metrics and code length, AST node count, cyclomatic complexity and cognitive complexity as static analysis metrics. Due to space limitations, we put detailed introduction to each metric in Table 6.

This framework rewards solutions that balance *functional correctness* with comprehensive aspects—a brute-force implementation passing all tests would score highly in correctness but poorly in complexity metrics, incentivizing refinement toward optimal algorithms.

## 3.3. Self-Critic & Process Rewarding

The same model $\mathcal{M}$ serves dual roles: generating reasoning along with solution candidates, and judging their viability.

After executing $c_t^{(j)}$ to obtain $F_t^{(j)}$, the model produces a texual critique $k_t^{(j)}$ and a numerical process reward $\rho_t^{(j)}$:

$$(k_t^{(j)}, \rho_t^{(j)}) = \mathcal{M}(\mathcal{R}_{t-1} \oplus r_t^{(j)}, C_{t-1} \oplus c_t^{(j)}, F_t^{(j)}).$$

The process reward grounds *subjective evaluation* of the reasoning process by combining *objective execution metrics*. This hybrid scoring prevents reward hacking—a model cannot inflate rewards without corresponding improvements in verifiable execution outcomes. Candidates with high $\rho$ but low $\omega$ signal outcome-process mismatches, which is a critical failure mode in conventional process supervision.

### 3.4. Unifying Process and Outcome Supervision

The beam search mechanism proceeds with Top-K successor states using a weighted *step score*: $q_t = \alpha \rho_t + \beta \omega_t$. Where $\alpha + \beta = 1$ governs the trade-off between theoretical soundness ($\rho_t$) and empirical effectiveness ($\omega_t$). The framework balances functional correctness while preserving promising reasoning trajectories.

*This formulation generalizes conventional supervision paradigms*. When $\beta = 0$, the framework reduces to pure process supervision akin to mathematical reasoning approaches that prioritize stepwise correctness over answers during reasoning. Conversely, $\alpha = 0$ recovers outcome supervision's focus on final code quality, similar to Best-of-N sampling but on a tree. Our unified perspective reveals these as endpoints on a continuum of supervision strategies, with the proposed $\alpha/\beta$ balance enabling simultaneous optimization of reasoning and outcome solution quality.

The synergy emerges through bidirectional feedback: execution outcomes ground reasoning process by identifying discrepancies between intended and actual behavior (e.g., test failures revealing flawed base cases in recursive algorithms), while process rewards guide exploration toward algorithmically superior implementations (e.g., recognizing that memoization could transform an $O(2^n)$ brute-force solution into an $O(n)$ dynamic programming approach).

## 4. Experiments

Our experimental evaluation aims to address three key questions: (1) How effective is our framework compared to existing approaches? (2) How does each component of our framework contribute to the overall performance? (3) What insights can we gain about the relationship between reasoning quality and code generation?

### 4.1. Experimental Setup

**Datasets.** We evaluate on 3 programming benchmarks as shown in Table 1. LBPP is a recent complex programming

dataset manually curated by human experts with competitive programming experience. HumanEval and MBPP are popular code generation benchmarks but could be trivial for current LLMs (Matton et al., 2024). Moreover, a significant proportion of the data is leaked in multiple pre-training corpora (Riddell et al., 2024). To ensure reproducibility, we report our detailed hyperparameters in Appendix A, we also open-source all our code and scripts.

**Unit Tests.** Our framework utilizes unit tests to verify and profile code solutions. We use LLM to generate these unit tests, which are then employed in computing the outcome reward $\omega_t^{(j)}$. However, self-generated unit tests do not always effectively assess code quality. Consequently, some prior works (Zhong et al., 2024) directly utilize dataset-provided unit tests. To facilitate a fair comparison with related approaches, we also consider using unit tests provided by datasets. In Table 2, methods using unit tests from datasets are denoted by **(w/ T)**.

**Baselines.** We compare several strong baselines for code generation. For outcome supervision, Reflexion (Shinn et al., 2024) is a recent self-improvement approach that utilizes execution results to refine generated code. LDB (Zhong et al., 2024) extends this by incorporating debugger outputs, and intermediate variable values for iterative solution refinement. For test-time scaling, we implement Best-of-N sampling, which generates multiple solutions and selects the best one based on test outcomes. Since no existing process supervision methods have been designed specifically for code generation, we adapt a similar approach from mathematical reasoning (Luo et al., 2024) in comparison, which we include in our ablation studies.

### 4.2. Main Results

Table 2 shows the comparative results of our method and baselines, Figure 3 provides detailed multi-dimensional profiling of the performance of generated solutions with different methods.

Table 1: **Dataset Statistics.** Characteristics of the programming benchmarks used in evaluation.

| | LBPP | HumanEval | MBPP |
|---|---|---|---|
| | (Matton et al., 2024) | (Chen et al., 2021b) | (Austin et al., 2021) |
| # Test Problems | 162 | 164 | 257[†] |
| # Unit Tests | 5.1 | 6.5 | 3.0 |
| Solution Length[§] | 627 / 3039 | 169 / 622 | 130 / 589 |
| Contamination | New Dataset | 18.9%[‡] | 20.8%[‡] |
| Difficulty | **Competitive Programming** | Basic Functions | Basic Functions |
| Task Type | Algorithms | Func. Completion | Basic Prog. |

[†]From `sanitized` version; [‡]Contamination results reported from Riddell et al. (2024);
[§]Average/maximum characters in solution code.

Table 2: **Main Results on Code Generation Benchmarks. Pass@1**: solutions passing all test cases. **Tests**: average test cases passed. **Valid**: solutions that compile and execute. **Time**: relative execution time, compared to the standard solution. Best results are in **bold** and second-best are underlined, every metric is in percentage.

| Model/Method | LBPP (2024) | | | | HumanEval (2021b) | | | | MBPP (2021) | | | |
|---|---|---|---|---|---|---|---|---|---|---|---|---|
| | Pass@1↑ | Tests↑ | Valid↑ | Time↓ | Pass@1↑ | Tests↑ | Valid↑ | Time↓ | Pass@1↑ | Tests↑ | Valid↑ | Time↓ |
| **Llama-3.1-8B-Instruct (2024)** | | | | | | | | | | | | |
| CoT | 30.9 | 44.3 | 63.0 | 176.8 | 50.0 | 68.4 | 82.9 | 98.1 | 58.0 | 64.9 | 72.4 | 91.9 |
| Reflexion | 34.0 | 49.3 | 67.3 | 148.5 | 54.9 | 71.1 | 83.5 | 107.5 | 58.8 | 65.0 | 71.2 | 88.6 |
| LDB (w/ T) | 25.9 | 39.8 | 58.0 | 252.2 | 54.3 | 62.3 | 66.5 | 127.1 | 43.6 | 47.1 | 49.4 | 170.7 |
| BoN | 46.9 | 64.7 | 84.6 | 107.6 | 71.3 | 84.7 | 93.3 | 77.3 | 73.5 | 79.9 | 86.4 | 72.1 |
| ORPS | 45.9 | 66.9 | 88.5 | 99.1 | 70.3 | 87.5 | 96.2 | 65.8 | 71.8 | 78.2 | 84.3 | 84.5 |
| ORPS (w/ T) | **67.1** | **81.4** | **93.7** | **89.4** | **91.4** | **95.7** | **98.1** | **63.6** | **90.4** | **93.1** | **95.6** | **59.1** |
| **DeepSeek-Coder-7B-Instruct-v1.5 (2024)** | | | | | | | | | | | | |
| CoT | 32.7 | 45.9 | 67.3 | 160.1 | 65.9 | 78.2 | 85.4 | 86.9 | 69.3 | 75.0 | 80.9 | 77.7 |
| Reflexion | 25.9 | 41.9 | 63.0 | 153.0 | 63.4 | 77.1 | 86.6 | 101.0 | 68.9 | 74.4 | 80.2 | 74.2 |
| LDB (w/ T) | 31.5 | 45.7 | 61.7 | 206.2 | 74.4 | 80.0 | 81.7 | 85.6 | 61.1 | 64.0 | 66.1 | 98.3 |
| BoN | 49.4 | 63.9 | 80.2 | 123.4 | 73.8 | 88.1 | 94.5 | 64.1 | 74.3 | 80.2 | 86.8 | 68.9 |
| ORPS | 56.3 | 71.1 | 88.0 | 89.4 | 76.2 | 90.0 | 96.3 | 40.6 | 73.2 | 80.3 | 87.5 | 46.8 |
| ORPS (w/ T) | **63.7** | **80.8** | **96.9** | **74.4** | **95.7** | **98.0** | **99.4** | **31.8** | **93.0** | **94.7** | **96.1** | **34.2** |
| **Qwen-2.5-Coder-7B-Instruct (2024)** | | | | | | | | | | | | |
| CoT | 40.1 | 55.3 | 72.2 | 118.6 | 72.6 | 79.0 | 82.3 | 79.2 | 79.0 | 83.3 | 88.3 | 67.3 |
| Reflexion | 37.7 | 57.1 | 78.4 | 111.2 | 75.6 | 81.1 | 84.1 | 73.6 | 79.0 | 84.0 | 88.7 | 63.5 |
| LDB (w/ T) | 35.8 | 49.9 | 65.4 | 187.8 | 87.8 | 90.3 | 91.5 | 76.1 | 66.9 | 69.4 | 72.0 | 96.8 |
| BoN | 53.1 | 68.8 | 85.8 | 117.9 | 77.4 | 85.1 | 87.8 | 66.8 | 82.9 | 87.2 | 91.8 | 62.6 |
| ORPS | 59.9 | 75.7 | 92.0 | 84.1 | 79.9 | 91.6 | 96.3 | 48.3 | 76.7 | 82.4 | 88.3 | 68.0 |
| ORPS (w/ T) | **77.8** | **87.9** | **96.9** | **82.4** | **96.3** | **98.0** | **98.8** | **43.9** | **94.9** | **96.4** | **97.3** | **45.3** |
| **Qwen-2.5-Coder-14B-Instruct (2024)** | | | | | | | | | | | | |
| CoT | 53.7 | 63.9 | 77.2 | 119.2 | 82.9 | 88.5 | 90.2 | 76.6 | 84.0 | 87.4 | 91.1 | 67.5 |
| Reflexion | 60.5 | 70.5 | 82.1 | 113.3 | 83.5 | 89.9 | 92.7 | 68.8 | 83.3 | 87.2 | 91.1 | 66.0 |
| LDB (w/ T) | 51.9 | 62.9 | 75.3 | 225.2 | 89.6 | 92.0 | 92.7 | 140.5 | 72.4 | 74.6 | 76.3 | 149.7 |
| BoN | 61.7 | 74.9 | 90.7 | 115.6 | 87.8 | 93.9 | 95.7 | 58.8 | 81.7 | 86.4 | 91.1 | 58.4 |
| ORPS | 61.7 | 77.4 | 90.7 | 84.8 | 81.7 | 91.3 | 96.3 | **41.5** | 76.3 | 82.0 | 87.9 | 58.8 |
| ORPS (w/ T) | **85.8** | **90.7** | **95.7** | **64.2** | **97.0** | **98.5** | **99.4** | 43.8 | **95.3** | **96.9** | **98.1** | **41.0** |
| **GPT-4o-Mini (2024)** | | | | | | | | | | | | |
| CoT | 50.0 | 65.9 | 80.2 | 124.5 | 79.9 | 87.5 | 90.9 | 80.5 | 78.6 | 83.5 | 87.9 | 70.3 |
| Reflexion | 62.3 | 73.9 | 87.7 | 93.2 | 75.0 | 83.6 | 87.2 | 75.1 | 79.4 | 84.0 | 88.3 | 67.6 |
| LDB (w/ T) | 54.9 | 67.8 | 82.7 | 220.1 | 88.4 | 92.2 | 93.9 | 133.4 | 72.8 | 75.5 | 77.8 | 157.9 |
| BoN | 64.2 | 78.6 | 93.8 | 88.9 | 82.9 | 90.2 | 92.7 | 66.5 | 80.5 | 85.5 | 89.9 | 64.6 |
| ORPS | 67.9 | 81.2 | 94.4 | 81.5 | 84.8 | 92.7 | 96.3 | 57.5 | 80.2 | 86.0 | 91.8 | 64.7 |
| ORPS (w/ T) | **88.9** | **94.3** | **98.1** | **61.6** | **97.6** | **98.7** | **99.4** | **46.2** | **95.7** | **97.3** | **98.4** | **51.4** |

Our results indicate significant improvements in both correctness and code quality metrics, especially on harder benchmarks. Even a smaller model (Qwen 7B), when paired with our method, could surpass its larger variant (Qwen 14B) without our method, suggesting that *providing sufficient reasoning space can be more effective than solely scaling model parameters* - which is significantly more computationally expensive. This finding has important implications for practical applications where computational resources are limited.

When compared to other execution-feedback and outcome reward based methods like Reflexion and LDB, our approach consistently demonstrates superior performance regardless of test case access. This improvement stems from a fundamental difference in approach: while these outcome-based methods focus primarily on local information like resolving execution errors and reasoning in chain structure, our method provides LLMs with *broader reasoning space to reflect on higher-level aspects such as algorithm selection and problem properties by using process reward guided search*. For instance, LDB achieves 35.8% Pass@1 on LBPP with Qwen-7B with test case access, while our method reaches 77.8% under the same conditions.

Particularly noteworthy is the performance boost when models have access to gold unit tests from test datasets (without access to solutions). All models show drastic improvements on all metrics in this setting. For instance, Qwen-7B achieves 77.8% Pass@1 on LBPP and 96.3% on HumanEval

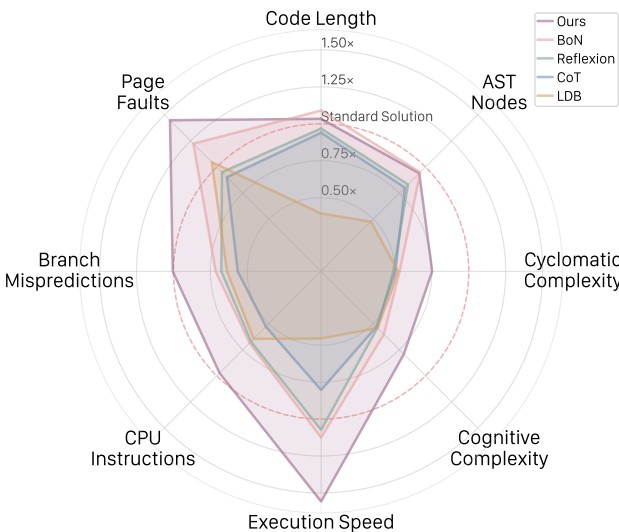

Figure 3: **Multi-dimensional Performance Analysis.** Metrics are normalized against the LBPP standard solutions (1.0×) and averaged across all backbone models. *Higher values indicate better performance.*

Table 3: **Ablation Study Results. - Execution**: Remove execution from our framework. **- Reasoning**: Remove reasoning process. Every metric is in percentage.

| Method | Pass@1↑ | Tests↑ | Valid↑ | Time↓ |
|---|---|---|---|---|
| ORPS | 59.9 | 75.7 | 92.0 | 84.1 |
| - Execution | 43.8 | 56.4 | 72.8 | 200.5 |
| - Reasoning | 55.6 | 74.5 | 94.4 | 124.5 |

with test case access, compared to 59.9% and 79.9% without. This suggests that while our self-generated test cases may be relatively weak, *given feedback for higher quality test cases, models can effectively guide themselves through the reasoning process to generate significantly better code.*

Figure 3 further supports these findings through detailed profiling results, showing consistent improvements over baselines across all models in terms of code efficiency and quality metrics. Guided by our framework, models are capable of refining themselves to generate faster, more coherent code. However, we do observe a slight disadvantage on MBPP, particularly when comparing with Best-of-N sampling. This is less concerning given that, as shown in Table 1, MBPP consists of relatively simple problems with short solutions, and a significant portion (20.8%) of its test data already exists in publicly available pre-training datasets.

### 4.3. Ablation Study

We conducted experiments on the challenging LBPP dataset using the Qwen-7B model to investigate the importance of two key components in the exploration process:

Table 4: **Analysis of Process Reward Model. Granularity** refers to the level of detail in the reward signal (line-level or outcome-level). **Train** indicates whether the process reward model requires training.

| Methods Granularity | Train | Pass@1↑ | Tests↑ | Valid↑ | Time↓ |
|---|---|---|---|---|---|
| Outcome | ✓ | 37.0 | 48.3 | 65.4 | 153.8 |
| Line | ✓ | 32.1 | 43.9 | 59.3 | 153.4 |
| Outcome | ✗ | 59.9 | 75.7 | 92.6 | 89.1 |
| Line | ✗ | 38.3 | 52.8 | 70.4 | 123.7 |

1. **Execution Outcomes.** Studies whether the model can access execution results to guide solution refinement. Without execution feedback, the model must rely solely on internal reasoning to assess correctness.

2. **Reasoning.** Investigates whether the model should explicitly perform reasoning before generating new code at each step. Without reasoning, the model directly generates new code based only on past code and execution results.

The results are presented in Table 3. When the model is unable to access execution outcomes during searching, Pass@1 decreases by 16.1%. *This highlights the critical role of environment feedback in guiding the model to generate correct solutions.* Since LLMs struggle to accurately predict execution outcomes for a given piece of code (Jain et al., 2024), incorporating execution results ensures that the model benefits from concrete feedback.

Omitting reasoning during searching results in a 4.3% decrease in Pass@1. *Reasoning enables the model to iteratively refine its approach based on feedback, addressing issues that may not be resolved through execution feedback.*

### 4.4. Analysis of Process Reward Model

Our framework challenges the necessity of training Process Reward Models (PRMs) by combining outcome rewards with reasoning to work as process rewards. While most prior work on process supervision (Luo et al., 2024) assumes that specially trained PRMs are required to guide reasoning during inference, we demonstrate that execution feedback as verifiable rewards, combined with existing LLMs' reasoning capabilities, can generate high-quality process rewards. This motivates us to investigate a fundamental question: *Is training PRMs necessary for effective code generation, or can outcomes alone provide sufficient supervision signals?*

**Experimental Design.** Table 4 presents a comparison across two dimensions: *granularity of supervision* (line-level vs. outcome-level) and *training approach* (trained vs. inference-only). For line-level methods, the model generates step-by-step reasoning with numerical rewards assigned

to each step, following mathematical reasoning approaches. For outcome-level methods, outcome rewards are computed after complete code generation and execution.

To determine whether the popular method of training line-level PRMs with higher-quality data can more effectively bridge the performance gap with our inference-only approach, we trained two PRMs with different data quality: `PRM-GPT` using a larger GPT-4o labeled dataset (containing 13644 synthetic labels for each line of reasoning text), and `PRM-Human` using human-filtered data where three authors spent 12 hours each to validate reasoning steps, achieving an inter-annotator agreement of 0.44 (Cohen's Kappa) and extracting 836 high-quality steps for training. This experiment thus directly tests if providing PRMs with optimal, high-quality training data enables them to match or surpass the performance of our method.

**Results and Analysis.** The results confirm several key insights. First, outcome-level reward signals prove more effective than line-level signals, as line-level feedback can only evaluate incomplete thought processes with limited information. Second, and more importantly, our inference-only approach substantially outperforms all trained PRMs given the same calls, even when PRMs are trained with high-quality human annotations.

When combined with the computational cost analysis (Table 5), these results indicate that while training data quality does impact PRM performance, trained PRMs still underperform our hybrid approach of verifiable execution rewards combined with LLM self-critique. This suggests that *outcomes alone, when properly integrated with LLM reasoning capabilities, provide superior supervision signals compared to learned reward models*. The effectiveness stems from grounding supervision in concrete, verifiable execution feedback rather than learned approximations.

### 4.5. Computational Cost Analysis

A critical concern raised by reviewers is whether ORPS's improvements come at the cost of increased computational overhead. To address this, we conduct controlled experiments limiting the total number of LLM calls, which is the bottleneck of inference-only methods, ensuring fair comparison under identical computational budgets.

**Complexity Analysis.** We analyzed the number of LLM calls by examining the source code of baseline methods. ORPS requires $2 \times N \times (K \times T + 1)$ calls, where $N$ is candidate samples, $K$ is beam size, and $T$ is reasoning steps. Reflexion (Shinn et al., 2024) requires $2 \times T$ calls, while LDB (Zhong et al., 2024) requires $1 + P \times T \times (2 + B \times N)$ calls, where $P$ is `pass_at_k`, $B$ is code blocks, and $N$ is max trials. For trained PRMs, we replace self-critic LLM calls with PRM calls, maintaining identical call counts.

Table 5: **Computational Cost Analysis.** Pass@1 performance on LBPP with Qwen-7B controlling LLM calls.

| Method | 20 Calls | 50 Calls | 100 Calls |
|---|---|---|---|
| Reflexion (2024) | 37.0 | 40.7 | 39.5 |
| LDB (2024) | 37.0 | 36.4 | 37.0 |
| REx (2024a) | 43.2 | 53.7 | 54.3 |
| PRM-GPT | 44.4 | 37.0 | 35.8 |
| PRM-Human | 40.7 | 38.3 | 42.0 |
| ORPS (**Ours**) | **48.4** | **55.6** | **64.2** |

**Results.** Table 5 presents results on LBPP with Qwen-7B under controlled computational budgets. ORPS consistently outperforms all baselines across different call limits, demonstrating superior computational efficiency. Notably, repair-focused methods like REx (Tang et al., 2024a) show diminishing returns as compute increases, while ORPS scales effectively. These results confirm that ORPS achieves better performance through more effective utilization of computational resources, not brute-force computation.

### 4.6. Scaling Analysis

While the previous analysis controlled LLM call budgets, we now examine performance scaling given the same number of generated code solutions. We compare ORPS against Best-of-N (BoN) sampling by varying candidate solutions on LBPP. The results in Figure 4 reveal distinct scaling behaviors: ORPS demonstrates superior scaling potential, with performance improving rapidly as more candidates become available. This suggests that *the structured reasoning in our framework more effectively identifies and refines promising solutions from the candidate pool*. In contrast, BoN exhibits slower improvements, indicating that simple selection from larger pools provides diminishing returns without strategic guidance.

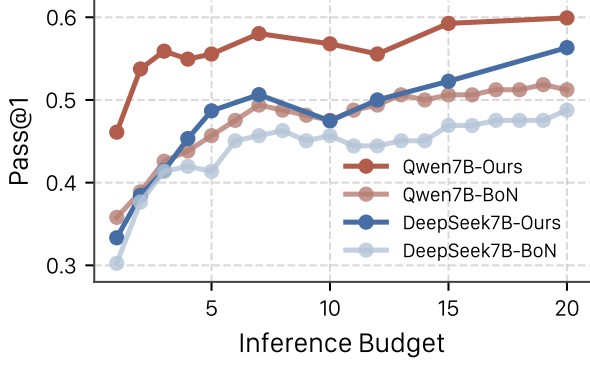

Figure 4: **Performance vs. Inference Budget.** The y-axis represents Pass@1 scores on LBPP. The x-axis represents the number of candidates generated during inference.

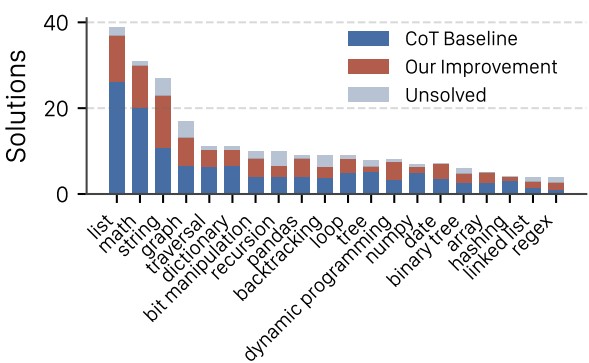

Figure 5: **Performance by Problem Class.** Top-20 problem classes in LBPP showing success rates and unsolved cases for ORPS vs baseline.

## 4.7. Case Studies

We also analyzed the improvements of ORPS across different problem categories. As shown in Figure 5, on the competitive programming dataset LBPP, our method shows significant improvements over the CoT Baseline, especially in more difficult categories. For instance, in complex algorithmic tasks such as dynamic programming, loops, and graphs, our method correctly solves nearly twice as many problems as CoT. This further confirms that *high-quality intrinsic reasoning can help models avoid logical pitfalls when tackling difficult coding tasks.*

Through detailed case studies, we demonstrate how our framework enhances code generation by improving reasoning. As shown in Appendix F, the response generated by the traditional CoT method for the *Minimum Greatest Common Divisor* problem in LBPP demonstrates that while the model provides a detailed thought process during solution generation, the complexity of the task results in an imperfect code implementation. For instance, in CoT's approach, the reliance on nested loops and pairwise GCD calculations introduces inefficiencies and fails to address scalability for larger datasets. Similarly, our method's initial implementation demonstrates a lack of robustness in handling edge cases and unnecessary redundancies in subset formation.

However, ORPS achieves a more accurate solution through finer reasoning. The code initially generated by our model contains redundancies and erroneous logic. Nevertheless, with the feedback from the critic on the execution outcomes, the programmer successfully refines the code to reach a correct implementation. *This iterative process not only eliminates logical errors but also optimizes performance, demonstrating the advantage of integrating structured feedback into code generation.*

## 5. Conclusion

We introduce Outcome-Refining Process Supervision (ORPS), a unified framework that fundamentally challenges the necessity of specially trained Process Reward Models in code generation. By combining execution-driven feedback with LLMs' intrinsic reasoning capabilities, ORPS demonstrates that verifiable outcomes can effectively guide process supervision without costly reward model training. Our comprehensive analysis across different data qualities—from GPT-4 labels to human annotations—shows that even high-quality trained PRMs underperform our inference-only approach, suggesting a paradigm shift in how we approach process supervision for complex reasoning tasks.

ORPS reveals several key insights: (1) **Strategic reasoning over local repair**: Unlike methods that incrementally fix code blocks, ORPS operates at a higher abstraction level, enabling exploration of fundamentally different algorithmic approaches rather than being trapped in local optima. (2) **Reasoning space over model scale**: Smaller models with ORPS often outperform larger variants using conventional methods, highlighting that structured reasoning frameworks can be more effective than pure parameter scaling. (3) **Computational efficiency**: ORPS achieves superior performance through strategic resource utilization rather than brute-force computation, scaling effectively while baselines show diminishing returns.

These findings collectively suggest that LLMs' intrinsic capabilities, when properly guided by verifiable signals, can eliminate the need for costly reward model training. Future work could explore extending this framework to other domains requiring rigorous reasoning and verification, opening new avenues for developing more efficient and practical reasoning frameworks in large language models.

## Impact Statement

This work advances code generation capabilities in large language models through more efficient reasoning processes. While our primary focus is methodological—improving algorithmic problem-solving without costly reward models—we acknowledge broader implications common to code generation systems. Enhanced programming assistants could democratize software development but may also lower barriers for generating malicious code. The framework's emphasis on code efficiency could reduce computational overhead in generated programs, though its environmental impact depends on deployment contexts.

These considerations reflect well-established societal trade-offs in AI-powered code tools rather than novel risks introduced by our approach. We encourage responsible deployment with standard safeguards against misuse, consistent with ethical practices for generative AI systems.

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

# A. Experimental Setup and Hyperparameter Details

This appendix provides a comprehensive description of the experimental setup, encompassing the hyperparameters, software, and hardware configurations employed in this study.

### A.1. Search Algorithm Hyperparameters (ORPS)

The following hyperparameters were used for the search algorithm in ORPS:

- **Search Depth (`num_rounds`):** 5. This parameter defines the maximum depth of the search tree, representing the number of iterative steps in the search process.

- **Beam Width (`top_k`):** 3. This parameter specifies the number of highest-scoring candidate solutions (traces) retained at each step of the beam search.

- **Expansion Factor (`num_samples`):** 20. This represents the number of new states (candidate solutions) explored from each state during the search process.

- **Process Reward Weight ($\alpha$):** 0.5. This metric determines the proportion of the Process Reward in the step reward.

- **Outcome Reward Weight ($\beta$):** 0.5. This metric determines the proportion of the Outcome Reward in the step reward.

Table 6: **Performance Metrics Description.** Our evaluation framework uses both dynamic execution profiling and static code analysis metrics to comprehensively assess code quality and efficiency.

| Category | Metric | Description |
|---|---|---|
| **Dynamic Execution Profiling** | | |
| | Time Enabled | Total CPU time spent executing the code, measured in nanoseconds. Lower values indicate more efficient execution and better algorithmic optimization. |
| | Instruction Count | Number of CPU instructions executed during runtime. Reflects computational efficiency, with lower counts suggesting more optimized code paths and better algorithm implementation. |
| | Branch Misses | Frequency of incorrect branch predictions during execution. Lower values indicate better code predictability and CPU pipeline efficiency, resulting in faster execution times. |
| | Page Faults | Number of times the program needs to access virtual memory. Fewer page faults suggest better memory management and more efficient memory access patterns. |
| **Static Analysis** | | |
| | Code Length | Total number of lines in the source code. Generally, shorter code length indicates more concise solutions while maintaining readability and functionality. |
| | AST Node Count | Number of nodes in the Abstract Syntax Tree. Measures structural complexity of the code, with fewer nodes suggesting simpler and more maintainable implementation. |
| | Cyclomatic Complexity | Quantifies the number of linearly independent paths through the code. Lower values indicate easier-to-maintain and test code, reducing potential bug sources. |
| | Cognitive Complexity | Measures how difficult the code is to understand, based on control flow structures and nesting. Lower scores suggest more readable and maintainable code that is easier to debug. |

## A.2. Inference Configuration

All inference experiments were conducted on a single machine using the FreeEval (Yu et al., 2024b) codebase, integrated with Hugging Face's `text-generation-inference` toolkit for efficient model serving. The following inference settings were applied:

- **Maximum Context Length (`max_tokens`):** 18,000 tokens. This parameter defines the maximum number of tokens allowed in the input sequence to the model.

- **Generated Tokens per Round:** 1,500 tokens. This specifies the number of new tokens generated by the model in each round of inference.

## A.3. Execution Constraints

To ensure consistent and reproducible results, the following execution constraints were enforced during inference:

- **Timeout per Test Case:** 5 seconds. This limits the maximum execution time allowed for each test case.

- **Memory Limit:** 512 MB. This constraint restricts the maximum memory allocation permitted for each test case.

- **Maximum Test Cases per Problem:** 15. This sets an upper bound on the number of test cases evaluated for each problem.

## A.4. Model Training Configuration

This section outlines the hyperparameters and settings used during the training phase of the model, which was pertinent to the analysis experiments (subsection 4.4). While ORPS itself does not require training, these details are provided for completeness and reproducibility.

- **Training Framework:** `llamafactory` (Zheng et al., 2024b)

- **Optimization Framework:** DeepSpeed ZeRO3 (Rajbhandari et al., 2020) (Zero Redundancy Optimizer Stage 3). This enables efficient training of large models by partitioning optimizer states, gradients, and model parameters across data parallel processes.

- **Base Model:** `qwen-2.5-coder-7b-instruct`. This is the pre-trained language model upon which further training was conducted.

- **Batch Size per Device:** 2. This defines the number of training examples processed on each GPU before a gradient update step.

- **Gradient Accumulation Steps:** 4. This allows simulating a larger effective batch size by accumulating gradients over multiple forward and backward passes before updating model weights. The effective batch size is therefore 8 (2 per device * 4 steps).

- **Learning Rate:** $2 \times 10^{-5}$. This parameter controls the step size taken during gradient-based optimization.

- **Learning Rate Scheduler:** Cosine decay. This gradually reduces the learning rate over the course of training, following a cosine function.

- **Number of Training Epochs:** 2.0. This specifies the number of complete passes through the entire training dataset.

- **Maximum Sequence Length:** 16,384 tokens. This defines the maximum length of the input sequences during training.

- **Mixed Precision Training:** Enabled with `bf16` (Brain Floating Point 16-bit format). This accelerates training by performing some computations with reduced precision while maintaining model accuracy.

## A.5. Hardware Environment

All experiments were performed on NVIDIA A800 GPUs, each equipped with 80GB of GPU memory.

# B. Additional Results on CodeContests

During rebuttal, reviewers were interested in an additional competitive programming contest benchmark, CodeContests (Li et al., 2022b). Due to page limitation, we include our new results here in Table 7.

Table 7: Pass@1 performance on CodeContests with Qwen-7B controlling LLM calls at 100.

| Method | Pass@1 | Tests % | Valid % | Time (ms) |
|---|---|---|---|---|
| Reflexion | 8.48 | 14.53 | 27.27 | 52318 |
| LDB | 7.88 | 11.79 | 21.21 | 3350 |
| REx | 13.33 | 20.17 | 32.12 | 27791 |
| PRM-GPT | 4.94 | 8.64 | 17.28 | 3084 |
| PRM-Human | 9.88 | 15.04 | 23.46 | **1985** |
| ORPS (Ours) | **20.61** | **28.36** | **40.61** | 36824 |

# C. AI Usage in Code Development

During the development of ORPS and the design of its experiments, LLMs were employed to assist with coding. All AI-assisted code were reviewed and refined by the authors to ensure correctness and alignment with the research goals.

# D. Impact of Optimizing Different Metrics on Code Quality

In ORPS, we integrate multiple evaluation metrics to guide the model's reasoning and code generation. These metrics include both **static analysis** (e.g., AST nodes, cyclomatic complexity) and **dynamic execution profiling** (e.g., execution speed, CPU instruction count, branch mispredictions). To better understand their individual contributions, we conduct an ablation study where each metric is optimized in isolation.

## D.1. Experimental Setup

To isolate the effect of each metric, we conduct a series of ablation experiments. In each experiment, the model receives **rewards only from a single metric**, while all other evaluation criteria remain unchanged. Specifically, we consider the following setups:

- **+AST Nodes**: Encourages structurally simpler code by minimizing the number of AST nodes.

- **+Cyclomatic Complexity**: Penalizes excessive branching and loop structures to improve maintainability.

- **+Cognitive Complexity**: Rewards code that is easier to understand based on nested structures and control flow.

- **+Execution Speed**: Optimizes for faster execution while maintaining correctness.

- **+CPU Instructions**: Minimizes the number of CPU instructions executed.

- **+Branch Mispredictions**: Encourages predictability to improve processor efficiency.

- **+Page Faults**: Reduces memory access overhead for better performance.

- **+All Metrics**: Incorporates all the above metrics into a single optimization objective.

## D.2. Results and Discussion

As shown in Figure 6, optimizing for a single metric significantly improves performance in that specific aspect, yet comes at a severe cost to other dimensions of code quality. This phenomenon suggests that the model falls into a form of **local metric optimality**, where it overfits to the given reward signal while neglecting other critical properties of high-quality code.

More specifically, we observe that optimizing for **static analysis metrics** (e.g., AST nodes, cyclomatic complexity) often leads to a sharp decline in **dynamic execution metrics** (e.g., execution speed, CPU instructions). For instance, minimizing

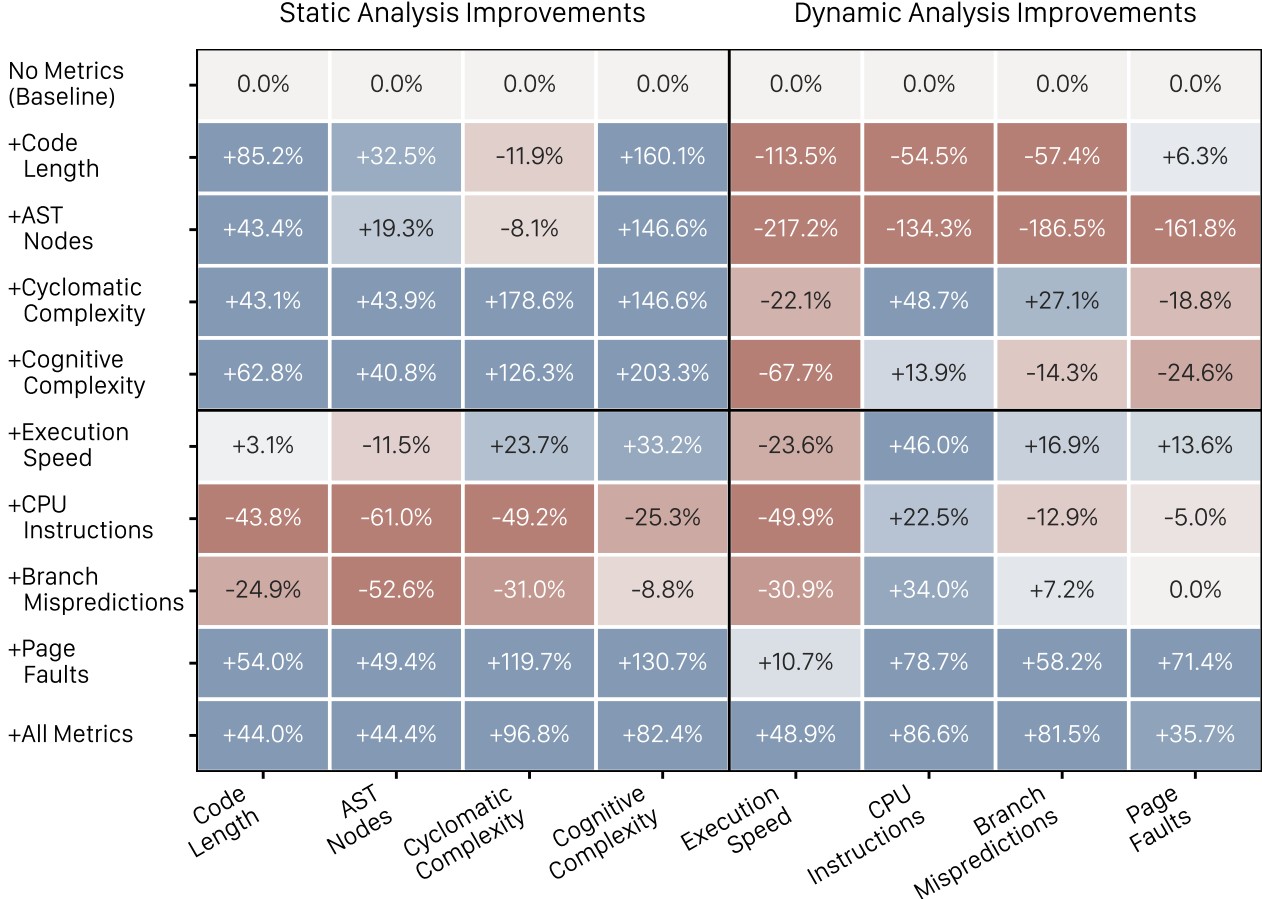

Figure 6: **Impact of single-metric optimization on code quality.** Each value represents the difference from the baseline. Optimizing for a single metric significantly improves performance in that dimension but leads to severe degradation in others.

cyclomatic complexity encourages structurally simpler code, yet it may suppress more efficient algorithmic choices that involve loops and conditionals. Conversely, optimizing for execution speed often results in obfuscated or redundant code, as the model prioritizes raw performance over maintainability.

These observations highlight an inherent challenge in code optimization: static structure and dynamic efficiency often conflict when optimized in isolation. This suggests that achieving well-balanced code quality requires a **multi-objective optimization** strategy rather than single-metric reinforcement. Traditional reward models struggle in such scenarios, as they often assume reward signals are aligned across different dimensions. However, our results indicate that code generation requires more nuanced supervision—one that dynamically balances trade-offs between readability, maintainability, and execution efficiency.

Furthermore, the steep performance drop in non-optimized metrics suggests that **reward sparsity** is a critical issue in single-metric training. Since the model receives no information about other quality dimensions, it fails to generalize improvements beyond the specific reward it observes. This contrasts with human programming intuition, where engineers naturally balance multiple objectives, such as runtime efficiency, readability, and modularity. Future work could explore techniques like **adaptive reward scaling**, where the model dynamically adjusts its focus based on real-time trade-offs rather than rigid metric-specific optimization.

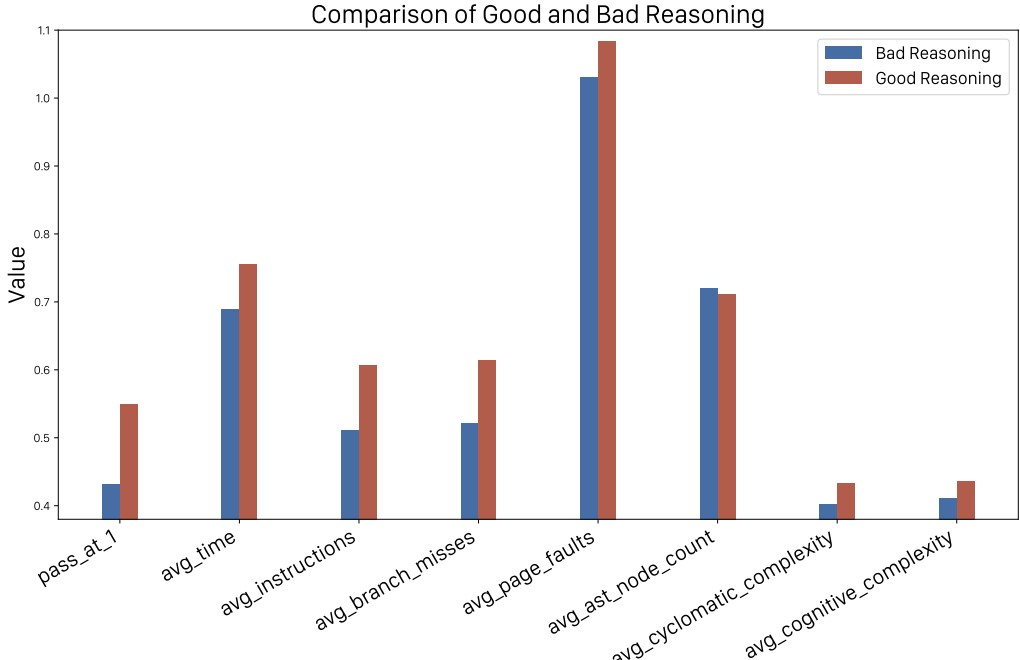

Figure 7: **Impact of reasoning quality on code quality.** Each metric is normalized, where higher values indicate better performance. The red bars represent the code quality when generated from higher-quality reasoning, while the blue bars correspond to lower-quality reasoning. The results suggest that better reasoning significantly improves execution-related metrics while having a limited effect on structural complexity.

## E. Correlation of Reasoning Quality and Code Quality

Our method is based on a core hypothesis: **higher-quality reasoning leads to higher-quality code**. To validate this assumption, we conducted a simple motivation experiment.

### E.1. Experimental Setup

We first prompted GPT-4o to generate two different reasoning chains for each problem in the LBPP dataset, explicitly ensuring a clear quality difference between them. To verify that the generated reasoning indeed exhibited significant quality differences, we used GPT-4o again to reassess and confirm their relative quality.

Next, we concatenated each reasoning chain with the original problem description and fed them separately into GPT-4o to generate code solutions. Finally, we evaluated the quality of the generated code across multiple metrics.

### E.2. Results and Analysis

As shown in Figure 7, all metrics are normalized, where higher values indicate better performance. The red bars represent the quality of code generated from higher-quality reasoning, while the blue bars correspond to the lower-quality reasoning.

From the results, we observe that **better reasoning generally leads to better overall code quality**, particularly in **dynamic execution metrics**. This suggests that high-quality reasoning not only improves correctness but also enhances execution efficiency, possibly by guiding the model to generate more optimal algorithmic structures.

Interestingly, the complexity of the generated code does not increase significantly. One possible explanation is that **more detailed and higher-quality reasoning naturally guides the model toward generating more sophisticated solutions that handle a wider range of scenarios**, rather than simply making the code structurally more complex.

These findings reinforce the importance of reasoning supervision in code generation. A strong reasoning framework helps models not only arrive at correct solutions but also optimize execution efficiency without unnecessarily inflating complexity.

# F. Example Model Outputs

To illustrate the effectiveness of our approach, we present a representative example from the LBPP dataset (Problem #101) that demonstrates how our method handles complex logical reasoning tasks. The original prompt of this problem is:

> You are given an array of integers, each of which is greater than 1. This array needs to be split into subsets such that the greatest common divisor within each subset is greater than 1. Write a Python program to return the minimum number of subsets required to create such a split.

---

**CoT Output**

Prob: Given an array of integers, split it into the minimum number of subsets such that the GCD of each subset is greater than 1.

Reasoning Process:

1. First, we need to understand what makes numbers group-able: - Numbers can be in the same subset if their GCD ¿ 1 - We want to minimize the number of subsets

2. Key observations: - If GCD(a,b) ¿ 1 and GCD(b,c) ¿ 1, a,b,c can potentially be in the same group - This suggests a graph-like structure where numbers are connected if their GCD ¿ 1

3. Solution approach: - Treat numbers as nodes in a graph - Connect numbers with edges if their GCD ¿ 1 - Find connected components in this graph - Each component represents one subset

4. Implementation strategy: - Use Union-Find data structure for efficient component tracking - Calculate GCD using Euclidean algorithm - Final answer is the number of connected components

Generated Solution:

```python
def gcd(a, b):
    while b:
        a, b = b, a % b
    return a

def minGroups(nums):
    n = len(nums)
    parent = list(range(n))

    def find(x):
        if parent[x] != x:
            parent[x] = find(parent[x])
        return parent[x]

    def union(x, y):
        parent[find(x)] = find(y)

    for i in range(n):
        for j in range(i + 1, n):
            if gcd(nums[i], nums[j]) > 1:
                union(i, j)

    return len(set(find(i) for i in range(n)))
```

## ORPS Programmer Output

\# === Programmer Thoughts === \# The problem requires splitting an array into subsets such that the greatest common divisor (GCD) within each subset is greater than 1. To achieve this, we can use a greedy algorithm. The idea is to iterate through the array and add each element to the existing subset if their GCD is greater than 1. If the GCD is 1, we start a new subset.

\# We need to implement a function that calculates the GCD of two numbers. Then, we can use this function to determine if an element can be added to an existing subset or if we need to start a new subset.

\# We also need to ensure that our solution is efficient. The time complexity of the GCD calculation can be optimized using the Euclidean algorithm, which has a time complexity of $O(\log(\min(a, b)))$.

\# === Solution ===

```python
import math

def gcd(a, b):
    """
    Calculate the greatest common divisor of two numbers using the Euclidean algorithm.
    """
    while b:
        a, b = b, a % b
    return a

def min_g_c_d_subsets(nums):
    """
    Split the array into subsets such that the greatest common divisor within
    each subset is greater than 1. Return the minimum number of subsets required.
    """
    subsets = []
    for num in nums:
        added = False
        for subset in subsets:
            if gcd(num, subset[-1]) > 1:
                subset.append(num)
                added = True
                break
        if not added:
            subsets.append([num])
    return len(subsets)
```

## ORPS Critic Feedback

\# === Critic Thoughts === The code provided attempts to solve the problem by iterating through each number and adding it to an existing subset if the GCD with the last element of the subset is greater than 1. If no such subset exists, a new subset is created. This approach ensures that each subset has a GCD greater than 1. However, there are a few issues and improvements that can be made:

1. **Test Case Failures**: The test cases for indices 5, 8, and 10 fail. This indicates that the current implementation does not correctly handle certain scenarios where numbers with no common divisor greater than 1 are present.

2. **Performance**: The code has a higher cognitive complexity, which means it is harder to understand and maintain. The use of nested loops and the nested if-else structure can be optimized.

3. **Efficiency**: The current approach has a time complexity of $O(n^2)$ due to the nested loops. This can be optimized to $O(n \log n)$ by sorting the numbers first and then using a single pass to form subsets.

4. **Code Clarity**: The code can be made more readable by breaking down the logic into smaller functions and adding comments.

5. **Handling Edge Cases**: The code does not handle edge cases such as an empty list or a list with a single element.

\# === Score === \$\$3\$\$

**ORPS Programmer Improvement**

# === Programmer Thoughts === # The feedback indicates that the current implementation has several issues, including failing test cases and high cognitive complexity. We need to address these issues by improving the logic and ensuring that all edge cases are handled correctly.

# **Key Improvements**: 1. **Handling Edge Cases**: Ensure that the function handles an empty list or a list with a single element. 2. **Efficiency**: Optimize the approach to achieve a time complexity of $O(n \log n)$ by sorting the numbers first and then using a single pass to form subsets. 3. **Code Clarity**: Break down the logic into smaller functions and add comments for better readability.

# **Approach**: 1. **Sort the Array**: Sorting the array ensures that numbers with a common divisor are adjacent, making it easier to form subsets. 2. **Single Pass to Form Subsets**: Iterate through the sorted array and add each number to the current subset if the GCD with the last element of the subset is greater than 1. If not, start a new subset. 3. **Edge Case Handling**: If the list is empty, return 0. If the list has a single element, return 1.

# === Solution ===

```python
import math

def gcd(a, b):
    """Calculate the greatest common divisor of two numbers
    using the Euclidean algorithm."""
    while b:
        a, b = b, a % b
    return a

def min_g_c_d_subsets(nums):
    """
    Split the array into subsets such that the greatest common divisor
    within each subset is greater than 1.
    Return the minimum number of subsets required.

    :param nums: List of integers greater than 1
    :return: Minimum number of subsets required
    """
    if not nums:
        return 0
    if len(nums) == 1:
        return 1

    # Sort the array to ensure numbers with a common divisor are adjacent
    nums.sort()

    subsets = []
    for num in nums:
        added = False
        for subset in subsets:
            if gcd(num, subset[-1]) > 1:
                subset.append(num)
                added = True
                break
        if not added:
            subsets.append([num])

    return len(subsets)
```

