# OpenReview forum: "Reasoning Through Execution: Unifying Process and Outcome Rewards for Code Generation"
_ICML.cc/2025/Conference — ICML 2025 poster_

### Official Review · Reviewer_4j6r · 2025-03-09

**Overall Recommendation:** 2

**Summary:**

This paper introduces Outcome-Refining Process Supervision (ORPS), a method unifying outcome supervision and process supervision in large language models (LLMs) for code generation tasks. The authors propose a tree-structured, inference-only, search framework using a combination of execution-based feedback, self-generated unit tests, and self-critiques to iteratively refine solutions. Empirical evaluations across three code-generation benchmarks (LBPP, HumanEval, MBPP) using different LLM models demonstrate improvements in correctness (Accuracy, or the Pass@1) and code efficiency metrics. The authors compare against LDB and Reflexion and show that the proposed ORPS achieves higher accuracy and more efficient code implementation.

**Claims And Evidence:**

The paper makes several claims, notably:

1. Structured reasoning guided by execution outcomes significantly improves code generation. Essentially, better textual reasoning leads to better code generation output.
2. A tree-structured search provides substantial performance gain due to multiple reasoning trajectories being maintained simultaneously.
3. Combining execution feedback with self-critique mechanisms creates a more reliable verification system than learned reward models. Notably, it showcases that a general model without PRM training is better than a trained PRM model.

Claims 1, 2 are well supported by experiments. Claim 3 is not convincing enough IMHO, despite empirical evidence given in Table 4 for the following reasons:
- The PRM training procedure is not clearly presented (from L366-370): It's not totally clear whether the line-level reward is collected by pure prompting GPT-4 to give line-level reward or it requires Monte-Carlo rollout, e.g. existing works such as [1] for math reasoning or [2] for code generation, and train another PRM with an modified value head.
- The PRM training is done on half of the LBPP dataset only, which contains 162 / 2 = 81 problem entries. It seems problematic to me to claim, solely from experiments on such small-scale training, that learned PRM is worse.

Moreover, Claims 1 and 2 are already undermined by existing literature:

1. The central claim that better reasoning improves outcomes has already been well-established by Olausson et al. [3], who demonstrated explicitly that reasoning quality and self-repair capabilities of LLMs correlate strongly with better outcomes in code generation. Notably, they also maintain a tree structure, called repair tree and show that the better quality of textual reasoning on previous failing code, such as coming from a stronger model or even human written, further increase the subsequent code generation performance.

2. Tree-structured search in LLM-based code refinement has been previously explored by Tang et al. [4] through a probabilistic exploration-exploitation approach for iterative code refinement, in which selecting which node in the tree to expand is heavily discussed.

Another minor claim that worth revisiting is the L161-163 and L68-73 saying that outcom-supervision could lead to inefficient solution such as brute-force solution: the correctness in these benchmarks is determined by unit tests of IO pairs w/ time and memory constraint; and could be avoided by adding in large IO pairs with tight timeout.

[1] OmegaPRM https://arxiv.org/abs/2406.06592

[2] Process Supervision-Guided Policy Optimization for Code Generation https://arxiv.org/abs/2410.17621

[3] Is Self-Repair a Silver Bullet for Code Generation? https://arxiv.org/abs/2306.09896

[4] Code Repair with LLMs gives an Exploration-Exploitation Tradeoff https://arxiv.org/abs/2405.17503

**Essential References Not Discussed:**

Please see above.

**Experimental Designs Or Analyses:**

The overall experiments look valid and sound, in particular for the main claim that ORPS is better than other variants such a LDB and Reflexion.

As I mentioned before, results in 4.4 are not convincing enough due to the lack of details of PRM and the small scale of PRM training being done. The presented Process Rewarding in Sec 3.3 and in L192 is not the same as Process Reward Model in Table 4, which creates confusion. The Process Rewarding proposed by the authors still operates on a full code snippet.

Also, the authors mentioned using generated unit tests in L246-247. There's no sufficient amount of details about how the authors generate the unit tests, the amount of the unit tests generated, and also a lack of reference for unit tests generation, which contains a rich literature either it being LLM generated or mutation methods.

**Methods And Evaluation Criteria:**

Benchmark:

- The authors report performance on 3 coding benchmarks, including LBPP, HumanEval, and MBPP. This is a legitimate choice, given the comparison is done against LDB https://arxiv.org/abs/2402.16906 primarily, which reports performance on HumanEval, MBPP. Still, it remains a question about whether the claimed evidence extends to competitive programming benchmarks such as CodeContests and LiveCodeBench. Also, the result could be more solid if the results are reported in an enhanced version benchmark, due to the false positive rate in the original ones, for example HumanEval+ or MBPP+.

Metrics:

- The main metrics reported are solely pass@1. Other code-specific metrics are also reported in Figure 3, which looks interesting. However, the pass@1 in the manuscript is closer to what is defined as "accuracy", due to the fact that it overlooks the number of code sampled. This has been discussed in [1] and [2], where both argue that pass@k should consider the number of code sampled in iterative refinement code generation process, or furthermore, a fairer metrics could be pass n@k used in [2, 3, 4] (for example a code generated followed by 2 self-repair attempts should be reported as pass 1@3 instead of pass@1 since 3 code snippets are generated) or pass@tokens in [1].

[1] Is Self-Repair a Silver Bullet for Code Generation? https://arxiv.org/abs/2306.09896

[2] What Makes Large Language Models Reason in (Multi-Turn) Code Generation? https://arxiv.org/abs/2410.08105

[3] Competition-Level Code Generation with AlphaCode https://arxiv.org/abs/2203.07814

[4] RLEF: Grounding Code LLMs in Execution Feedback with Reinforcement Learning https://arxiv.org/abs/2410.02089

**Other Comments Or Suggestions:**

Notations are not clear, some symbols are never defined:

- What is $j$ in L213?
- What is $m_k^{j}$ in the equation in L177?
- How is the weighted step score $q_t$ (L212) used? The beam size is only presented in Algo 1 and seems to be not mentioned in the main text.

**Other Strengths And Weaknesses:**

Strengths:

- A clear and practical inference-only framework that integrates execution-based feedback with LLM-driven critiques.
- Empirical results indicating meaningful improvements in both correctness and efficiency. The including of metrics other than pass rate looks interesting.

Weaknesses:

- There're non-negligible flaws in the claims and the methods part (Please see above). More analysis would be appreciated; for example, the number of LLM calls, or the number of tokens generated, or the number of invocations of code evaluation of ORPS compared to LDB & Reflexion for a fairer comparison rather than just "Accuracy".
- Limited conceptual novelty, as the core ideas (reasoning improvement from an LLM for self-repair, tree search) were previously established. Otherwise clarification from the authors is highly appreciated.
- Presentation and writing could have been improved. Some important details are missing such as unit tests generations.

**Questions For Authors:**

Please see above.

**Relation To Broader Scientific Literature:**

Please see above.

**Theoretical Claims:**

There's no theoretical contribution in the manuscript, as it's a paper with empirical findings.

---

> ### Author Rebuttal · Authors · 2025-03-28
>
> Thank you for your review. We address your concerns point by point:
> ## Metrics
> Unlike methods that report Pass@k which attempts k solutions on testset (e.g. LDB and Self-Repair use ground-truth test cases which can be considered contamination), **our Pass@1 measures attempting only 1 solution on testset**. We only generate and execute multiple codes during reasoning on self-generated test cases. The final code (one per problem) is evaluated on the actual test set (unless marked "w/ T"). In fact, although baselines like LDB report Pass@k where k solutions are tested against testset cases, they sample much more code and debug on test cases from testset to obtain each solution. Thus it's still a fair comparison if we control the attempts on the full testset cases for all methods. Furthermore, we report metrics beyond Pass@1 as shown in Table 5.
>
> ## PRM Training
>
> We report training details in Appendix A.4. While we used 81 problems for PRM training to avoid contamination, we collected a larger dataset by beam search with 20 reasoning steps each. This generated 5414 steps with GPT-4o labels for training. **To further address your concern on label quality, we trained 2 new line-level PRMs:**
>
> 1. PRM-GPT: A larger GPT-4o labeled dataset (13644 steps).
> 2. PRM-Human: To obtain better reward labels, we sampled 400 reasoning chains with GPT labels and ask 3 authors to spend 12hrs each, to classify if the chains and rewards are valid. We got an average inter-annotator agreement of 0.44(Cohen's Kappa,AB=0.27,AC=0.61,BC=0.44). We kept 209 valid chains from 261 chains that annotators agree on each other and extracted 836steps for PRM training.
>
> ## Cost
> We add controlled experiments on LBPP with Qwen7B limiting LLM calls(Details in Rebuttal DEgP) with 2 new PRMs and REx[4].
>
> |20 Calls|Pass@1|Tests%|Valid%|Time%|
> |-|-|-|-|-|
> |Reflexion|37.0|51.7|71.6|119.5|
> |LDB|37.0|50.8|66.0|274.7|
> |REx|43.2|57.4|72.2|268.4|
> |PRM-GPT|44.4|58.1|77.8|**100.1**|
> |PRM-Human|40.7|53.1|69.1|124.7|
> |ORPS|**48.4**|**64.8**|**84.5**|105.6|
>
> |50 Calls|Pass@1|Tests%|Valid%|Time%|
> |-|-|-|-|-|
> |Reflexion|40.7|57.6|79.6|130.3|
> |LDB|36.4|50.4|66.0|272.6|
> |REx|53.7|66.6|84.0|199.9|
> |PRM-GPT|37.0|52.9|71.6|**112.7**|
> |PRM-Human|38.3|53.5|69.1|137.7|
> |ORPS|**55.6**|**72.1**|**89.5**|116.8|
>
> |100 Calls|Pass@1|Tests%|Valid%|Time%|
> |-|-|-|-|-|
> |Reflexion|39.5|54.4|72.8|113.8|
> |LDB|37.0|51.0|66.7|275.7|
> |REx|54.3|65.6|79.0|218.4|
> |PRM-GPT|35.8|51.1|70.4|127.3|
> |PRM-Human|42.0|56.4|76.5|106.6|
> |ORPS|**64.2**|**75.4**|**88.9**|**91.0**|
>
> Results show repair-based methods (LDB, Reflexion) fail to scale effectively. LDB focuses on fixing code blocks in a single solution, which works for simple bugs but fails when the algorithmic approach is suboptimal, while ORPS improves significantly. REx may appear advantaged as it generates code every call while others generate code every 2 calls, attempting twice as many solutions given the same calls. **Given the same compute budget, ORPS consistently generates better solutions without training or test cases from testset.**
>
> ## Novelty and Contributions
> ORPS fundamentally challenges the assumption in process supervision(OmegaPRM,MathShepherd) that specially trained PRMs are necessary for reasoning guidance. Compared to trained PRMs, although increasing training data quality(which is expensive) could lead to improvements, **combining verifiable rewards with existing LLMs without training outperforms trained PRMs as compute increases**. This is significant conceptual advancement for process supervision that eliminates the overhead of PRM training while improving results.
>
> Unlike Self-Repair, REx, LDB that focus on repairing code or specific blocks, ORPS reasons about solution strategies at a higher level to avoid local optima. Instead of incrementally fixing specific solutions to pass more tests, ORPS explores different algorithms with extensive reasoning, even on solutions that pass all generated cases for improvement.
>
> Compared to REx which focus on selecting best nodes to explore by solving arm-acquiring bandit problem from exec outcomes, we propose **LLMs are capable of directly generating high quality process rewards given some feedback to select and expand certain reasoning chains**. In fact REx might be used in conjunction with ORPS process rewards since REx focus on selecting better nodes while we try to produce better rewards but is out of the scope of this work.
>
> Besides, previous methods require testset cases, which is practically hard to obtain(as pointed out by DEgP and LiveCodeBench), while ORPS relies on self-generated cases(**for unit-test generation, please refer to Rebuttal DEgP**).
>
> For notation issues, we will fix them in revised version. We will also cite, discuss and compare with works you listed that we missed previously,like REx and Self-Repair.
>
> **Given these clarifications and new results demonstrating ORPS's superior performance, we respectfully and strongly request reconsideration of your score.**

---

> > ### Comment · Reviewer_4j6r · 2025-04-04
> >
> > Thank you for the detailed response and the clarification of the novelty.
> >
> > My concerns in my original reviews contain mainly 2 parts: 1. a fair comparison of the budget and report performance beyond "Accuracy" 2. Novelty and contributions could have been clearer if the manuscript is better contextualized given the existing literature.
> >
> > The results with controlled LLM calls, and addition results for comparison of PRM and REx strength the claim of the superiority of ORPS compared to the other methods and address my concern 1.
> >
> > The other flaws, such as notation and wording, are in my eye somehow "fixable"; I thus encourage the authors to incorporate the suggestions during the discussion, and the manuscript could benefit from the enhanced clarify for presentation. In particular, the PRM part already confuse me and Reviewer DEgP.
> >
> > For concern 2, I see the novelty as the **combination** of the techniques presented such as unit test generations, tree search, and reasoning on the previous attempt and execution, instead of individually, as some of them overlap with the findings in the literature (c.f. the ref in my original comment) already.
> >
> > > Unlike Self-Repair, REx, LDB that focus on repairing code or specific blocks, ORPS reasons about solution strategies at a higher level to avoid local optima. Instead of incrementally fixing specific solutions to pass more tests, ORPS explores different algorithms with extensive reasoning, even on solutions that pass all generated cases for improvement.
> >
> > This claim is very interesting: I agree with the authors that algorithmic reasoning and self-repair (to fix runtime or wrong answer error, to achieve better perf on HumanEval or MBPP) are different and the compared method could be "myopic" in this sense. And the authors could have been centered on this for the presentation of the manuscript and the claim could benefit more if eval is done on some commonly-used competitive programming benchmark such as CodeContests, TACO, or LiveCodeBench, along with the given existing  results to show the difference.
> >
> > Taking these into account, I bump my score.

---

> > > ### Author Response · Authors · 2025-04-05
> > >
> > > We sincerely thank you for your thoughtful and constructive comments, which have significantly contributed to improving our work.
> > >
> > >
> > > > the claim could benefit more if eval is done on some commonly-used competitive programming benchmark such as CodeContests, TACO, or LiveCodeBench, along with the given existing results to show the difference.
> > >
> > > Thank you for suggesting additional competitive programming benchmarks to validate our claims.
> > >
> > > Regarding our original selection of datasets, we select MBPP and HumanEval as these are the most commonly used code generation benchmarks. We chose LBPP because it explicitly addresses data contamination and ensures difficulty by asking human experts with competitive programming experience to manually curate and verify problems from scratch.
> > >
> > > **Following your suggestion, we've now added experiments on CodeContests from DeepMind (test split) with all baselines using the same compute budget (100 LLM calls) on Qwen 2.5 Coder 7B.** Note the time metric is not normalized since CodeContests do not provide standard Python solutions for each problem thus we report average running time.
> > >
> > > | 100 Calls | Pass@1 | Tests % | Valid % | Time (ms) |
> > > |-------|--------|--------------|----------------|-----------|
> > > | Reflexion | 8.48 | 14.53 | 27.27 | 52318 |
> > > | LDB | 7.88 | 11.79 | 21.21 | 3350 |
> > > | REx | 13.33 | 20.17 | 32.12 | 27791 |
> > > | PRM-GPT | 4.94 | 8.64 | 17.28 | 3084 |
> > > | PRM-Human | 9.88 | 15.04 | 23.46 | **1985** |
> > > | ORPS | **20.61** | **28.36** | **40.61** | 36824 |
> > >
> > > These results align with our previous findings, further confirms that our approach scales effectively on complex tasks that require higher-level algorithmic thinking rather than mere code repair.
> > >
> > >
> > > Regarding novelty, we appreciate your recognition of our core contribution. Moreover, one of our core insight that **"combining verifiable rewards with existing LLMs without training outperforms trained PRMs as compute increases"** represents a significant advancement. This direction has been validated by several preprints after the ICML deadline (e.g. S∗: Test-Time Scaling for Code Generation [arXiv:2502.14382] mentioned by Reviewer DEgP, and a more recent one Review, Refine, Repeat: Dynamic Evaluation and Selection [arXiv:2504.01931]), confirming that we identified a promising research direction.
> > >
> > >
> > > We also thank you for pointing out presentation issues (the "fixable" flaws). We will definitely improve the clarity (including more details and the focus), and fix all notation issues. We'll also incorporate all the suggestions from the reviewers and our rebuttals.
> > >
> > > Your thoughtful feedback have been invaluable to us. They have helped us refine our empirical evaluation and pushed us to articulate our contributions more precisely. We are genuinely grateful for the time and expertise you've invested in reviewing our work. Given the additional evidence now supporting our claims and your recognition of our contribution's significance, we respectfully ask for your consideration in further raising your score of our paper.

---

### Official Review · Reviewer_xFyb · 2025-03-11

**Overall Recommendation:** 4

**Summary:**

The paper proposes outcome refining process supervision (ORPS), a unified framework to bridge the gap between process supervision and outcome supervision through structured reasoning and execution-based feedback. The experiment results show that concrete feedback signals are pivotal for solving complex programming tasks. Furthermore, the work shows that eliminating trained process reward models with hybrid process rewards significantly boosts model performances.

**Claims And Evidence:**

Yes, the paper provided evidences to support the claims made in the work. For example, the paper establishes: code generation offers a unique opportunity through concrete, verifiable signals. The paper further shows that eliminating the need for specially trained process reward models help in improving reasoning quality.

**Essential References Not Discussed:**

I didn't carefully check but there seems to be enough paper in the reference.

**Experimental Designs Or Analyses:**

The paper performed experiments on three popular programming problem solving benchmarks. The evaluation setup is well detailed and looks sound. Quite a few strong models are evaluated with the proposed approach.

**Methods And Evaluation Criteria:**

The proposed method is composed of three steps, namely (1) candidate code generation, (2) executing candidate codes and run profiling on unit tests, and (3) self-critic and process rewarding. All these 3 steps make sense for the code generation. The evaluation is performed on three popular code generation benchmarks.

**Other Comments Or Suggestions:**

None.

**Other Strengths And Weaknesses:**

The paper is overall well written. The paper has done rigorous experiments which is the strength of the work.

**Questions For Authors:**

1. Does the proposed approach composed of components which already exists in the literature? If yes, how do the authors see the novelty of the work?
2. I am a bit confused but wanted to confirm, during inference, does the proposed approach iterative generation? I mean does it generate and then refine the generation?
3. If I understand correct, the proposed method does not require any training, right? It is an inference-only approach?

**Relation To Broader Scientific Literature:**

It seems like the contributions made in this paper is not highly novel, if we compare to the existing literature. For example, self criticism is not a new idea. Getting execution feedback to get an accurate criticism is not new as well. However, if we judge the proposed method as a whole, it seems quite interesting and effective.

**Theoretical Claims:**

The paper does not have any proofs or theoretical claims.

---

> ### Author Rebuttal · Authors · 2025-03-29
>
> Thank you for your positive review and recognition of our work's effectiveness. We appreciate your thoughtful questions and address them below.
>
> >  **Q1:** Does the proposed approach composed of components which already exists in the literature? If yes, how do the authors see the novelty of the work?
>
> Regarding novelty, while we integrate some established concepts, we introduce several key innovations rather than merely combining existing components:
>
> 1. Prior process supervision approaches (OmegaPRM, Math-Shepherd, etc.) assume specially trained Process Reward Models are required for effective reasoning guidance. **ORPS challenges this assumption and studies necessity of training PRMs, by combining execution feedback as verifiable rewards with existing LLMs to guide reasoning.** Results show that ORPS, as a pure inference-time method, produces better reasoning guidance than trained PRMs.
> 2. Unlike repair-based approaches (e.g., Self-Repair, LDB) that incrementally fix specific code blocks, ORPS reasons at a higher abstraction level about solution strategies. This enables exploring different algorithms rather than being trapped in local optima when the initial approach is suboptimal. Moreover, unlike previous works, we also aim to generate code solutions that are correct, efficient, easy to understand and maintain, instead of focusing solely on correctness.
> 3. ORPS effectively operates without access to ground truth test cases from benchmark testsets. ORPS utilizes self-generated unit tests during reasoning - a practical advantage in real-world scenarios where test cases are unavailable.
>
> >  **Q2:** I am a bit confused but wanted to confirm, during inference, does the proposed approach iterative generation? I mean does it generate and then refine the generation?
>
> Yes, ORPS is an iterative generation approach. It maintains multiple solution trajectories simultaneously through beam search, where each iteration involves reasoning, code generation, execution, and self-critique. This iterative process allows for strategic pivots when initial approaches prove suboptimal.
>
> > **Q3:** If I understand correct, the proposed method does not require any training, right? It is an inference-only approach?
>
> Correct, ORPS requires no training whatsoever. It's a fully inference-time framework that leverages an LLM's existing capabilities for reasoning, code generation, and self-critique. Our experiments in Table 4 and additional analyses (in our response to Reviewer 4j6r) confirm that this inference-only approach outperforms methods requiring PRM training, especially as compute budget increases.
>
> Thank you again for your supportive review. We believe ORPS represents a significant step forward in unifying process and outcome supervision for complex code generation tasks.

---

> > ### Comment · Reviewer_xFyb · 2025-04-04
> >
> > Thank you for addressing my questions.

---

### Official Review · Reviewer_Kbe7 · 2025-03-14

**Overall Recommendation:** 4

**Summary:**

- The paper proposes an LLM-based algorithm (outcome-refining process supervision, ORPS) for code generation. The primary contributions are algorithmic and empirical. ORPS leverages LLMs abilities significantly more than prior work in this area. Specifically, this includes self-reflection, self-critique and process (per-step) rewards. The key difference from prior LLM-based codegen models is that ORPS eliminates the need for a separately trained process reward model (PRM), which provides dense execution feedback at each step of the iterative reasoning chain. Combined with tree search, this leads to significant performance improvement on modern codegen benchmark datasets.

**Claims And Evidence:**

- Yes.

**Essential References Not Discussed:**

- No (but I'm not certain).

**Experimental Designs Or Analyses:**

- Not applicable.

**Methods And Evaluation Criteria:**

- Yes.

**Other Comments Or Suggestions:**

- No additional coments.

**Other Strengths And Weaknesses:**

Strengths
  - The paper tackles an important and relevant problem. Improvement here is likely to have a large impact and be of large interest to the community.
  - The paper is well structured and mostly clearly written. The main ideas are easy to follow and nicely illustrated.
  - The experiments show significant performance improvement compared to strong baselines.


Weaknesses
  - Overall, the paper is quite good. I wasn't able to spot any major technical issues. My biggest concern is around reproducibility. I didn't see the full prompt or prompts used as inputs to the LLM listed in any of the appendices. The format of the LLM responses "Programmer Thoughts", "Critic Thoughts", and numerical rewards suggest some prompt engineering was likely involved. If yes, they should be included in the appendices to aid reproducibility. (Please note that my current rating of the paper assumes a clear listing of the system prompts, if any, used to interact with the LLM will be included in the author response.)

  - The paper can be somewhat handwavy with terminology and some claims. A description or definition of concepts and terms before their usage would be preferable. Examples include "reasoning", "reasoning space", "finer reasoning", "intrinsic reasoning", and so on.

**Questions For Authors:**

- (Q1) What initial LLM inputs (system prompts) used in each call to the LLM, if any. If system prompts exist, is it possible to include them in the appendix.

- (Q2) What's the current SOTA for LBPP? Is it ORPS?

**Relation To Broader Scientific Literature:**

- The paper extends the state-of-the-art (to my knowledge) of using LLMs for code generation. It builds on very recent prior work (LDB, Reflexion). It leverages the LLM significantly more. That said, LLMs-for-code-generation is a very fast moving area so I can't be certain that I haven't missed relevant prior work or baselines.

**Theoretical Claims:**

- Not applicable.

---

> ### Author Rebuttal · Authors · 2025-03-29
>
> Thank you sincerely for your thoughtful review and constructive feedback. We deeply appreciate your recognition of our work and we address your questions and concerns point by point:
>
> > **Q1:** What initial LLM inputs (system prompts) used in each call to the LLM, if any. If system prompts exist, is it possible to include them in the appendix.
>
> We sincerely appreciate your emphasis on reproducibility. **Reproducibility is our first priority** and thus we have anonymously open-sourced all implementation details, including code, prompts, setup instructions, commands to run and reproduce our experiments in our anonymous repository. The repository contains all prompts used in ORPS. We attach our prompts at the end of this response. However, due to space constraints of rebuttal, we cannot paste everything here, but **we will make sure to include all prompts in our revised appendix**. For the new experiments and data involved in rebuttal, we will also add them to our repository.
>
> > **Q2:** What's the current SOTA for LBPP? Is it ORPS?
>
> To the best of our knowledge, ORPS currently achieves SOTA performance on the LBPP benchmark. Moreover, this improvement requires no additional model training and can be applied to any sufficiently capable base LLM. As demonstrated in our experiments, ORPS consistently outperforms other baselines across multiple models given the same compute budget.
>
> > **W2:** The paper can be somewhat handwavy with terminology and some claims. A description or definition of concepts and terms before their usage would be preferable. Examples include "reasoning", "reasoning space", "finer reasoning", "intrinsic reasoning", and so on.
>
> Thank you for pointing out our presentation issues. We acknowledge the need for clearer definitions and will revise the paper to explicitly clarify key terms and provide more explanation to our notations. We will ensure all technical terms are clearly defined before their first use.
>
> We are very grateful for your expertise and time in evaluating our work. Thank you again for your contributions to improving this work.
>
> The system prompt of model for reasoning and generating codes:
>
> ````
> You are a Python programmer in a code interview. You're solving a coding problem while sharing your thoughts and discussing with an critic. Always follow the format below for each round:
> 1. First, share your thoughts as comments:
>    - Your current approach (or an analysis of the problem and your plan if you haven't written any code yet)
>    - What you learned from previous feedback (if there is any previous feedback, otherwise think about your plan, what might be missing from your previous plan)
>    - Why you chose this approach (or how you plan to tackle the problem), do you need to shift your approach?
>    - Be clear, concise, detailed and pay attention to the comments given by the critic, no chit-chat, no small talk.
>    - Always use first person, e.g. I, we, our, etc.
> 2. Then write your solution:
>    - Clean, efficient Python code that follows requirements exactly
>    - No test cases in code, just the solution
>    - Your code will then be tested by the critic, so do not include any test cases in your code, this is very important
> Format your response as:
> # === BEGIN PROGRAMMER THOUGHTS ===
> # [Your response to previous feedback]
> # === END PROGRAMMER THOUGHTS ===
> # === BEGIN SOLUTION ===
> ```python
> [Your code]
> ```
> # === END SOLUTION ===
> Output strictly as shown above.
> ````
>
> The system prompt for the self-critic role:
>
> ```
> You are a technical critic reviewing a programmer's Python code. Provide clear, constructive feedback and suggest improvements or propose a new approach. The programmer's thoughts, code, execution analysis will be provided to you each round and you need to give constructive feedback to the programmer.
> Here are some rules you need to follow:
> 1. First, share your thoughts:
>    - How did the code perform in the execution, did any test cases fail?
>         - If any case failed, why it failed? If all passed, think about performance improvement.
>    - Key observations on the code, e.g. what's good or bad, what's the most important thing to improve, etc.
>         - Potential improvements: time / space complexity, a different algorithm, strategy, etc.
>    - Think about the programmer's thoughts, propose a new direction if you have any better idea, or give some advice to the plan, or give additional analysis of the problem
>    - Your thought process should be clear and concise, no chit-chat, no small talk. Guiding the programmer to improve their code is your main goal, you do not write any code.
>   - Always use first person, e.g. I, we, our, etc.
>
> ...
> <omitted due to character limitation of rebuttal, please refer to our open-source repository for details>
> ```

---

### Official Review · Reviewer_DEgP · 2025-03-14

**Overall Recommendation:** 2

**Summary:**

This paper introduces ORPS, a novel framework that unifies outcome and process supervision to address complex code problems. Notably, this approach does not require training PRMs. ORPS demonstrates significant improvements when utilizing ground truth test cases.

**Claims And Evidence:**

Overall, most of the claims are well-claimed. Please check my other comments below.

**Essential References Not Discussed:**

N.A.

**Experimental Designs Or Analyses:**

This paper should include a more detailed cost analysis of both the dynamic and static analysis processes.

**Methods And Evaluation Criteria:**

The method lacks persuasiveness because the salient improvement assumes the availability of ground truth test cases, which is uncommon in real-world code completion scenarios. Without these ground truth test cases, the performance advantage is not prominent.

**Other Comments Or Suggestions:**

N.A.

**Other Strengths And Weaknesses:**

The main weakness of the paper is that the performance improvement heavily depends on the availability of ground truth test cases, which is uncommon in real-world scenarios.

Additionally, the rationale for combining dynamic and static analysis is unclear. For instance, the necessity of static analysis is questionable when dynamic analysis has already been performed. If static analysis is indeed necessary, there should be more consideration regarding the weighting of scores between static and dynamic analysis.

**Questions For Authors:**

1. It's somewhat puzzling that the paper begins with Process Reward Models. I initially assumed the designed reward would be used to train the model, but it's not addressed in this paper. Instead, it seems more like an inference-time strategy, similar in spirit to [1]. It's important to note that a minor resemblance to [1] does not affect the paper's rating. My question is why the narrative starts with Process Reward Models rather than an inference-time strategy when the paper doesn’t involve training. Please note that this affects the understanding of contribution 2 to eliminate specially trained PRMs

2. Additionally, while I understand that a tree-structured exploration space could enhance the pass rate of generation, I'm curious why the instances mentioned in this paper focus solely on efficiency, which is a non-functional requirement, rather than on functional aspects measured by the pass rate. This is evident in the instances mentioned in lines 72-73 and 161-163.



[1] S∗: Test Time Scaling for Code Generation

**Relation To Broader Scientific Literature:**

The primary contribution of this paper is the integration of dynamic and static analysis to construct the reward.

**Theoretical Claims:**

N.A

---

> ### Author Rebuttal · Authors · 2025-03-28
>
> Thank you for your valuable feedback.
>
> ## Answers to Questions
>
> **Q1**: Currently, there's an assumption in process supervision and test-time scaling research (e.g. OmegaPRM, Math-Shepherd, Let's Verify Step by Step, Deepseek-Math, etc.) that a specially trained Process Reward Model is required to guide reasoning (during training with RL or inference with search algorithms like MCTS). **ORPS fundamentally challenges this assumption and study necessity of training PRMs, by combining execution feedback as verifiable rewards with existing LLMs to generate high quality process rewards to guide reasoning.** We compare ORPS with trained PRMs by using different methods to guide LLM reasoning during inference (Table 4) and we now add 2 new experiments (in rebuttal for 4j6r) to extensively validate our claims.  Results indicates ORPS produce better rewards than trained PRMs to guide reasoning and consistently performs better given the same compute budget.
>
> Regarding S*[1], this work was made public one month after the ICML submission deadline which further validates our approach.
>
> **Q2**: We agree that the certain phrasing suggest a focus on efficiency over correctness. To clarify: our framework prioritizes optimizing correctness as the primary objective, while considering comprehensive metrics involving efficiency, code quality, complexity, etc. In all our experiments we report correctness metrics along with efficiency for comprehensive comparison. **Our motive here is that a good solution should be correct, efficient (evaluated with dynamic analysis), easy to read and maintain (static analysis). Furthermore, we study how dynamic and static analysis metrics affects overall performance(Figure 6 in Appendix D) and this should also address Weakness 2.**
>
> ## Unit Test Cases
>
> Thank you for raising the concern on the scarcity of ground truth test cases in practical scenarios. We agree quality of test cases might significantly affect the performance of all approaches that utilize execution feedback to debug or improve the code. Our baseline Reflexion points out "they rely upon ground truth test cases that invalidate pass@1." **However, we have explicitly addressed this concern, and the test cases used in ORPS are not the ground truth from test sets.** Instead, we follow Reflexion to prompt the LLM given the problem and example tests and filter invalid cases. These self-generated, weak tests are then used during reasoning, and only one final solution will be tested against the actual ground truth cases. To ensure fairness and reproducibility, we cached the cases generated by Qwen 7B and used the same cases in all our experiments. **While the quality of generated cases are suboptimal compared to ground truths, ORPS performs better, even compared to the baselines using ground truth cases.** Notably, we also evaluate ORPS with ground truth test cases (ORPS (w/T) in Table 1) in comparison. We will be including all system prompts in revised Appendix.
>
> ## Computational Cost
>
> The bottleneck of computational cost of ORPS and baselines is LLM Inference Calls instead of dynamic and static analysis, as this require GPU computing and introduce the majority of latency. Although we compared ORPS with BoN in Figure 4 by allocating the same number of samples per problem, we now include a new experiment for other baselines given the same number of LLM calls.
>
> To be precise, we analyze the worst case cost for each method: ORPS requires $2\times N \times (K \times T + 1)$ Calls, where N = Candidate Samples, K = Beam Size, T = Reasoning Steps. Reflexion requires $2\times T$ calls, where T = Steps. LDB requires $1 + P×T×(2+B×N)$ where P = pass_at_k, T = max_iterations, B = number of blocks in code control flow graph, N = max_trials. For trained PRMs, we swap the self-critic LLM call with the PRM and thus the calls are identical. Please refer to rebuttal for 4j6r for PRM training details.
>
> |20 Calls|Pass@1|Tests%|Valid%|Time%|
> |-|-|-|-|-|
> |Reflexion|37.0|51.7|71.6|119.5|
> |LDB|37.0|50.8|66.0|274.7|
> |PRM-GPT|44.4|58.1|77.8|**100.1**|
> |PRM-Human|40.7|53.1|69.1|124.7|
> |ORPS|**48.4**|**64.8**|**84.5**|105.6|
>
> |50 Calls|Pass@1|Tests%|Valid%|Time%|
> |-|-|-|-|-|
> |Reflexion|40.7|57.6|79.6|130.3|
> |LDB|36.4|50.4|66.0|272.6|
> |PRM-GPT|37.0|52.9|71.6|**112.7**|
> |PRM-Human|38.3|53.5|69.1|137.7|
> |ORPS|**55.6**|**72.1**|**89.5**|116.8|
>
> |100 Calls|Pass@1|Tests%|Valid%|Time%|
> |-|-|-|-|-|
> |Reflexion|39.5|54.4|72.8|113.8|
> |LDB|37.0|51.0|66.7|275.7|
> |PRM-GPT|35.8|51.1|70.4|127.3|
> |PRM-Human|42.0|56.4|76.5|106.6|
> |ORPS|**64.2**|**75.4**|**88.9**|**91.0**|
>
> Results indicate that **even we use weaker, self-generated test cases, ORPS consistently outperform baselines that use ground truth test cases given the same compute budget and scales up consistently.**
>
> We sincerely appreciate your time and constructive feedback, which has helped us strengthen our work. **Given these these clarifications and new results, we respectfully request reconsideration of your score.**

---

### Decision · Program_Chairs · 2025-05-01

**Decision:**

Accept (poster)

**Comment:**

This paper presents an inference-time methodology for improved accuracy in code generation. Like a lot of work in this space it assumes access to test cases. It achieves state of the art results across different base models. Reviewers are broadly positive, but the one weak objector takes issue with the requirement of ground truth test cases. While this is an important limitation, it does not a limitation that is specific do this paper but one which is specific to this entire family of works.

Therefore I recommend accepting the paper, but I suggest that the authors revise to discuss all of the related work that the reviewers raised: This is a crowded and fast moving area, so I don't fault the authors not discussing everything, but they should spruce up the related work as described by the reviewers.